SciPost Physics

# Translation Groups for arbitrary Gauge Fields in Synthetic Crystals with real hopping amplitudes

**Marco Marciani**

Dipartimento di Fisica, University of Naples "Federico II", I-80126 Naples, Italy

## Abstract

The Cayley-crystals introduced in [F. R. Lux and E. Prodan, Annales Henri Poincaré 25(8), 3563 (2024)] are a class of lattices endowed with a Hamiltonian whose translation group $G$ is generic and possibly non-commutative. We show that these systems naturally realize the generalization of the so-called magnetic translation groups to arbitrary discrete gauge groups. A one-body dynamics emulates that of a particle carrying a superposition of charges, each coupled to distinct static gauge-field configuration. The possible types of gauge fields are determined by the irreducible representations of the commutator subgroup $C \subset G$, while the Wilson-loop configurations – which need not be homogeneous – are fixed by the embedding of $C$ in $G$. The role of other subgroups in shaping both the lattice geometry and the dynamics is analyze in depth assuming $C$ finite. We discuss a theorem of direct engineering relevance that, for any cyclic gauge group, yields all compatible translation groups. We then construct two-dimensional examples of Cayley-crystals equivalent to square lattices threaded by inhomogeneous magnetic fluxes. Importantly, Cayley-crystals can be realized with only real hopping amplitudes and in scalable geometries that can fit higher-than-3D dynamics, enabling experimental exploration and eventual exploitation in metamaterials, cQED, and other synthetic platforms.

# 1 Introduction

A uniform magnetic field threading a crystal enlarges the effective electronic unit cell by a factor determined by the magnetic flux through the original cell in units of the flux quantum [1, 2]. Such phenomenon is a consequence of a symmetry reduction. The Hamiltonian still commutes with the so-called magnetic translation operators – which equal the translation operators up to a phase – but they fail to form a group representation of $\mathbb{Z}^n$ (for some dimension $n$ of the crystal) [3]. Instead, they form a "weaker" projective representation. It turns out that these projective representations can be lifted to be a proper group representation if they can act on a larger group that covers $\mathbb{Z}^n$ [4]. Such groups are the so-called magnetic translation groups (MTGs) [5, 6].

From a mathematical point of view, MTGs have been described as certain *central extensions* of the translation group by the gauge group $U(1)$ [7], or $\mathbb{Z}_m$ as a discrete version. Such extension problems have been tackled in finite or infinite crystals by some authors [8–13] resulting in an algorithmic recipe for working out all cohomologically inequivalent, but possibly isomorphic [14], MTGs. Although generalizations of magnetic operators have been developed thoroughly in a number of settings, e.g. in aperiodic [15] or disordered [16] lattices, and with advanced algebraic tools [17], generalizations of MTGs have been mostly focused on fundamental physics and continuous systems [7, 18, 19].

In this paper, we will discuss in which cases the generalization of MTGs to arbitrary discrete gauge groups and to inhomogeneous periodic field patterns is mathematically possible. Moreover, we identify a conceptually simple system called Cayley-crystals as a promising candidate for an experimental implementation of such groups, where their properties can be tested and exploited.

Cayley-crystals have been introduced in Ref. [20] and consist of one-body Hamiltonians whose translation group could be a representation of any abstract discrete group $G$. The primary reason for the introduction of these systems was to understand spectral properties of the so-called hyperbolic lattices [21–24], ruled by the non-amenable and infinite Fuchsian groups [25], through the asymptotics of their finite quotients. In their simplest realization (one degree of freedom per site), such Hamiltonians describe the dynamics of a quantum particle over the set of vertices of the Cayley-graph of $G$ generated by the set of hoppings.

Cayley-crystals seem to be the right object to achieve the generalization of the MTG to all other *gauge translation groups* (GTGs). In fact, the group $G$ itself of the Cayley-crystal can be understood as such a GTG. In particular, the so-called *commutator* subgroup $C$ of $G$ defines the discrete gauge group. Crucially, $G$ is an extension of $\mathbb{Z}^n$ but not necessarily a central one, enabling the generalization from homogeneous field patterns to inhomogeneous ones. The fact that a Cayley-crystal embodies a GTG has an impact on the dynamics. Any $N$-dimensional irreducible unitary representation (irrep) of $C$ identifies a charge sector of the quantum particle and each charge bears a gauge configuration (abelian and magnetic only if $N = 1$) as in a gauge theory with frozen dynamics of the gauge fields [26].

We shall discuss some engineering aspects. First of all, to allow easy computations and to relate to experimentally feasible systems, we shall confine ourselves to the study of finite gauge groups $C$. The associated Cayley-crystals can still be infinite but, as we shall show, have finite periodic unit cells and different types of periodic boundary conditions (PBCs) can be implemented. Secondly, to make evident the analogy between Cayley-crystals and systems bearing gauge fields, we shall work with a specific arrangement of the Cayley-graph which we refer to as the Cayley-Schreier-lattice (CS-lattice). Vertices corresponding to the same commutator cosets are arranged linearly on top of a $n$-dimensional hypercubic lattice, thus forming a $(n + 1)$-dimensional system where the gauge field acts locally. Moreover, by a simple (quasi-isometric) embedding, such systems can be made three-dimensional – and thus experimentally

realizable – without giving up the gauge locality. Due to their inherent scalability, CS-lattices can be engineered in synthetic systems or metamaterials where hoppings can be guided via some mesa as is done typically in cQED setups [27–32] and can be envisioned in waveguide arrays [33, 34] or mechanical devices [35, 36]. Importantly, gauge fields can emerge even in CS-lattices with real hopping terms. This property is highly desirable, as complex phases typically add complexity in experimental setups [37] and require fine-tuning across the lattice to impress specific gauge fluxes. Finally, Ref. [38] presents a theorem producing a canonical way to construct all possible CS-lattices starting from $C$ as the desired gauge group. We shall discuss here a simpler version that assumes $C$ cyclic, which is enough to demonstrate many properties of the GTGs and the dynamics they can cover.

The paper is organized as follows. In Section 2 we briefly define Cayley-crystals associated with generic discrete groups. In Section 3 we introduce the reader to CS-lattices. The mapping of the particle dynamics in a CS-crystal to that in a Euclidean lattice with gauge fields is done in Section 4. The problem of constructing GTGs from a given gauge group is tackled in Section 5. To deepen the analysis, we study the periodicity of the crystals and uncover the existence of a number of different unit cells, which trivialize in Euclidean crystals, in Section 6. Implementations of different types of PBCs are discussed in Section 7. By means of such unit cells we refine the spectral analysis in Section 8. Combining all previous results, we present and discuss all CS-crystals with $C = \mathbb{Z}_2, \mathbb{Z}_3$ in Section 9. General considerations and a comparison with the current literature can be found in Section 10. In Appendix A we show how to perform the 3D embedding of a CS-lattice while the other appendices concern less important details.

## 2  Cayley-crystals

Cayley-crystals are lattices equipped with a Hamiltonian $\mathcal{H}$ that is invariant under a (possibly) non-abelian translation group $G$ [20]. They generalize standard Euclidean crystals, where $G = \mathbb{Z}^n$ (for some dimension $n$) and is abelian. When $G$ is non-commutative, its commutator subgroup [39]

$$C = \left\langle c_{g_1 g_2} = \bar{g}_1 \bar{g}_2 g_1 g_2 \mid g_1, g_2 \in G \right\rangle$$

is nontrivial. Here, each $c_{g_1 g_2}$ is called a *commutator*, and we denote inversion by an overbar on group elements. We shall assume $C$ finite, since in experimental applications any lattice is finite, and possibly non-abelian.

The Hamiltonian is a Hermitian operator acting on square-integrable wavefunctions of the form

$$|\psi\rangle = \sum_{g \in G} \psi(g)|g\rangle \tag{1}$$

where the set $\{|g\rangle\}_{g \in G}$ forms the standard orthonormal basis of $L^2(G)$. Translations in $L^2(G)$ are described by two operators called right(left) regular representations $\mathcal{T}^{R(L)}$ where $\langle g|\mathcal{T}_\lambda^R|\psi\rangle = \psi(g\lambda)$ and $\langle g|\mathcal{T}_\lambda^L|\psi\rangle = \psi(\bar{\lambda}g)$ ($\lambda, g \in G$). They encode translations since their action is *transitive*, that is, a wavefunction localized at a site $g$ can be entirely displaced at $g'$ by the action of one element of either representations, and *free*, that is, there are no fixed points for non-trivial translations. In Euclidean crystals $\mathcal{T}^R = (\mathcal{T}^L)^{-1}$; for a non-abelian $G$ the two representations have a slightly more complex relation, but they keep commuting.

Since $\mathcal{T}_G^{R(L)}$ is isomorphic to $G$, by definition $\mathcal{H}$ must commute with one of the two representations. Without loss of generality, we impose

$$[\mathcal{H}, \mathcal{T}^R] = 0.$$

As pointed out in Ref. [20] this commutation is naturally possible if the operators in $\mathcal{H}$ are left regular representation, that is,

$$\mathcal{H} = \sum_{\lambda \in \Lambda} \kappa_\lambda \, \mathcal{T}_\lambda^L \tag{2}$$

with $\Lambda \subseteq G$ a subset of hoppings with associated coefficients satisfying $\kappa_\lambda = \kappa_{\bar{\lambda}}^*$. For physical reasons that will become clear later, we shall assume $\Lambda$ to be a generating set.

## 3 Cayley-Schreier-lattices

The most natural setup that physically implements the Hamiltonian of Eq. (2) has arguably the structure of a Cayley-graph. The vertices of this graph are the elements of $G$ while the edges are placed between the vertices connected by the group elements in $\Lambda$. We observe that the most natural implementation of $\mathcal{T}^L$ is through *real* permutation matrices that describe the vertices connections. Then, if the $\kappa_\lambda$ are real, so is the Hamiltonian.

Cayley-graphs can be drawn in many ways and we seek those that emphasize the structure of the commutator subgroup as an inherently local feature. To this end, we shall add a prescription about the placement of the vertices, which is the content of this section.

### 3.1 A convenient Cartesian coordinates system

All elements of $G$ can be (coset-)decomposed in a unique way as

$$g = g^{ab} c \tag{3}$$

where $c \in C$ and $g^{ab}$ is in a transversal set in $G$ in bijection with $G^{ab} = G/C$. $G^{ab}$ is called the *abelianization* of $G$ and, by the fundamental theorem of abelian groups, is isomorphic to $\mathbb{Z}^n \times X_{\tilde{n}}$, with $X_{\tilde{n}}$ a product of $\tilde{n}$ finite cyclic groups [39]. For simplicity we assume $\tilde{n} = 0$, the other cases are briefly commented in Section 5 and do not bring conceptual novelty.

To consistently fix the decomposition Eq. (3), we choose the so-called Schreier transversal $T_n^{ab}$ [40] so that $g^{ab} = g_{\mathbf{r}} = \prod_i^n x_i^{r_i} = x_1^{r_1} x_2^{r_2} \dots x_n^{r_n}$ for some powers $r_i = \mathbf{r}^{(i)}$ of the minimal generators $x_i$ of $G$ – it is natural to choose $x_i \in \Lambda$ but not necessary. The transversal is not a group since a product $g_{\mathbf{r}_1} g_{\mathbf{r}_2} = g_{\mathbf{r}_1 + \mathbf{r}_2} h(\mathbf{r}_1, \mathbf{r}_2)$ might not be in $T_n^{ab}$. In particular, the so-called 2-cocycles $h(\mathbf{r}_1, \mathbf{r}_2) \in C$ can be quite an ugly product of commutators.

Thus, from a set-theoretic point of view, $G$ is equivalent to the Cartesian product $C \times T_n^{ab}$ and Eq. (3) produces a satisfactory coordinate system for the group elements. Moreover, it suggests a natural embedding of the Cayley-graph of $G$ in $\mathbb{R}^{n+1}$, as we describe in the following.

First, we may arrange the elements of $T_n^{ab} \subset G$ according to their label $\mathbf{r}$ to form a translation-invariant hypercubic $n$-dimensional lattice, which we shall call *abelian support*. Then, since $C$ is finite, its elements $c_i$ can be ordered and arranged on line above each element of $T_n^{ab}$ in the additional dimension, to form what we shall call *pillars* (see an example with $n = 2$ in Fig. 1a). In Appendix A, we discuss that it is always possible to embed such a support into a two-dimensional (2D) plane to have an overall 3D implementation. Finally, we equip the lattice with edges that connect elements that differ by elements in $\Lambda$. We assumed the latter to be a generating set of $G$ otherwise the lattice splits into disconnected components. Thus, elements pertaining to different pillars are connected. Connection within the pillars are possible if $\Lambda \cup C$ is nontrivial but, for graphical simplicity, we shall assume in the figures and examples that this is not the case. We observe that connectivity depends on whether $s \in \Lambda$ is added to the right or left of the starting element. We name such graphs Cayley-Schreier (CS)-lattices (see Fig. 1b) but, in the following, we shall consider only lattices obtained from left multiplication since the other type describes isomorphic graphs.

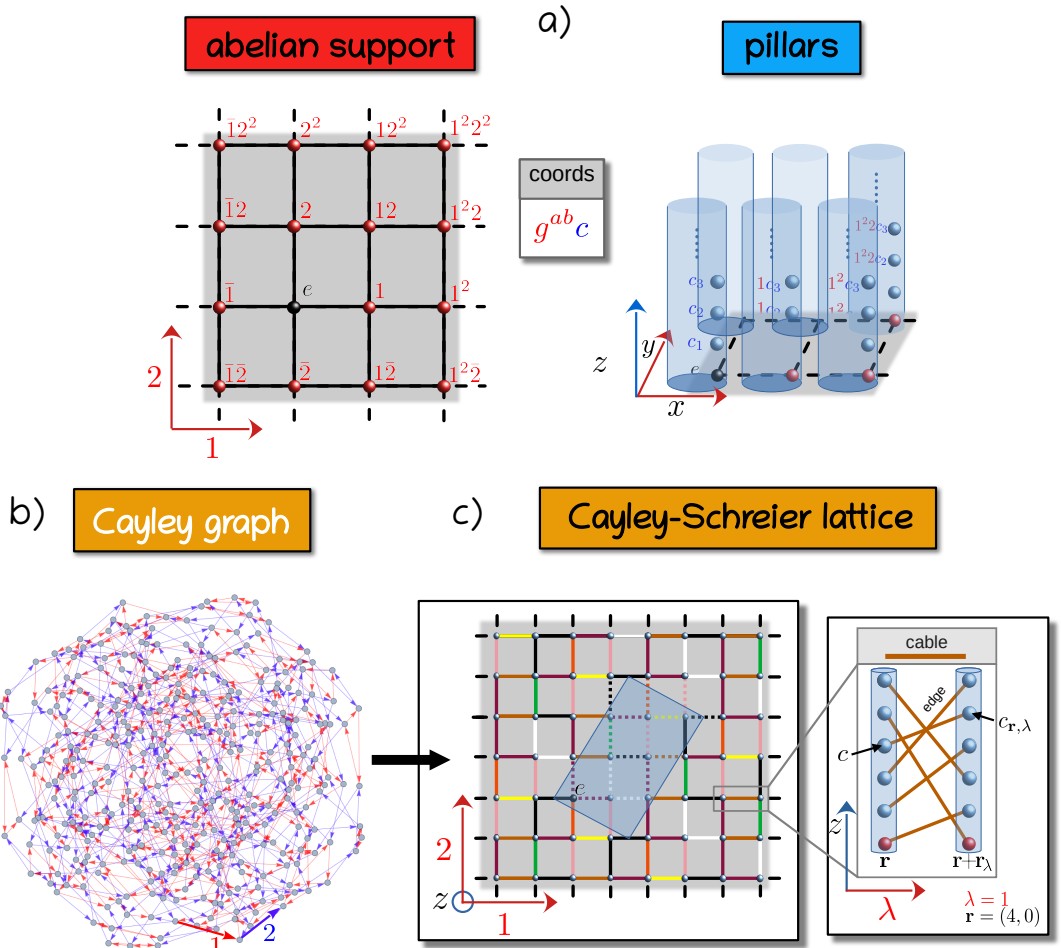

Figure 1: (a) Cayley-Schreier-lattice structure. On the left, abelian support $T^{ab}_{n=2}$. On the right, side view of the pillars on the abelian support. We shall often short the generators notation with $x_i \to i$. (b) Example of Cayley-graph for the alternating group $A_6$ (of order 360) to be compared with CS-lattice type of graph. (c) Sketch of a CS-lattice. The periodic standard unit CS-cell is shaded in blue, the cables pertaining to it have been depicted as dashed lines. On the side, structure of a cable. The sketch is made to highlights a general case of tilted cells that might appear with $n > 2$ – here $n = 2$ and this specific pattern violates Corollary E.4, realistic patterns are shown in Section 9) – and boundary cables are not shown.

This grouping of the cosets of $C$ into pillars that are local on the abelian support bears a resemblance to Schreier coset-graphs [41], where cosets of the Cayley-graph are collapsed to single points and the edges are defined by generators in $\Lambda$. Whence the chosen name for the lattice. To make this parallelism even stronger, we may bunch all edges flowing from one pillar to another into a single bundle of lines that we shall call *cable* (see the zoom inset in Fig. 1c). Using cables instead of edges and imagining to collapse pillars into a single artificial atoms with as many levels as the pillar elements, we obtain graphically almost a Schreier coset-graphs. However, the atoms will be connected not simply via the generators but via elements of a subgroup of the symmetric group $S_n$, which are matrices that permutes the levels and represent the commutator group. The mathematical details are given in the next paragraphs.

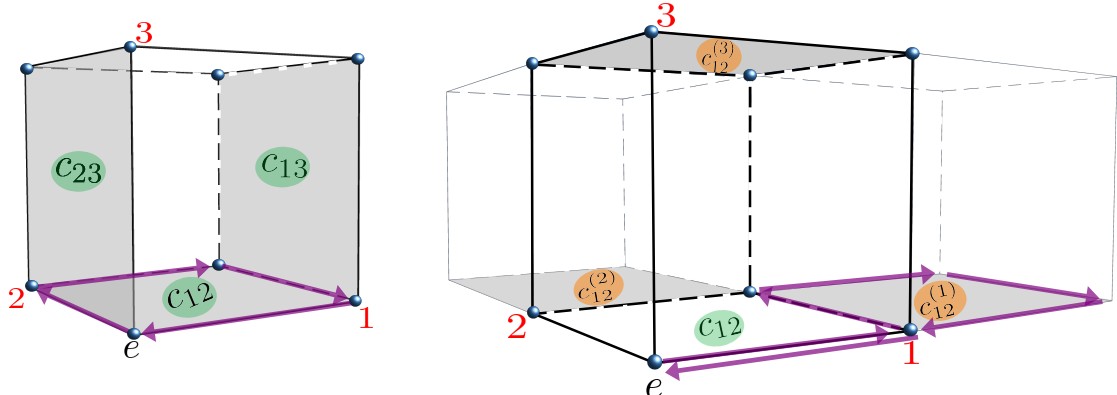

Figure 2: (left) Cubic cell in a 3D abelian support located at the origin. The three *parent* commutators are marked on their corresponding faces. The loop associated to $c_{12}$ is shown with purple arrows that implement, from right to left, the sequence $\bar{1}\bar{2}12$. (right) Cubic cells adjacent to the origin. *Adjacent* commutators are marked on their corresponding faces (only those relative to $c_{12}$ are shown). The loop sequence associated to $c_{12}^{(1)}$ is shown.

## 3.2 Links and cables on the CS-lattices

We define the *link* in the direction $\lambda \in \Lambda$ starting from a pillar at point $\mathbf{r}$ of the abelian support to be the quantity

$$l_{\mathbf{r},\lambda} = \overline{g_{\mathbf{r}+\mathbf{r}_\lambda}}\, \lambda\, g_{\mathbf{r}} \in C \tag{4}$$

In Proposition 1.1 and Corollary 1.2 we give explicit expressions of links in terms of products of elementary elements of the commutator group. These formulas are handy since in Section 5 we will show that presentations of infinite CS-lattices without boundary conditions have relators expressed only in terms such elementary constituents.

We notice that $l_{\mathbf{r}_2,g_{\mathbf{r}_1}}$ coincides with the 2-cocycles $h(\mathbf{r}_1,\mathbf{r}_2)$; therefore, they are important geometric quantities. In particular they determine the *connection field* of the lattice and describe in a discrete context how to "parallel" transport the elements of a pillar at a certain position under left multiplication by a lattice generator $\lambda$, via the relation

$$c_{\mathbf{r},\lambda}(c) = l_{\mathbf{r},\lambda}\, c \tag{5}$$

where $c$ is a pillar element at position $\mathbf{r}$ and $c_{\mathbf{r},\lambda}(c)$ a pillar element at position $\mathbf{r} + \mathbf{r}_\lambda$.

The function $c_{\mathbf{r},\lambda}$ describes the cables introduced in the above paragraph. Since links are $C$-elements, at most $|C|$ different types of cables appear in a CS-lattice.

We shall see in Section 4 that links and cables are also relevant analytical objects, as they appear in the representations of $G$ induced by $C$ when decomposing the Hamiltonian along the different gauge sectors.

## 3.3 Commutators and Wilson-loops

Hereafter, we denote conjugated $C$-elements as $c^{(g)} = \bar{g}\, c\, g$ ($g \in G$) and simplify the notation for the generators with $x_i \rightarrow i$.

We shall refer to elements $c_{ij}$ as *parent* commutators and to elements $c_{ij}^{(g_{\mathbf{r}})}$ as *adjacent* commutators. Graphically, they correspond to elements of the pillar above the identity element.

Clearly, they can be accessed starting from the identity element and following one by one the links, respectively, of the loop at the origin around face 'ij' and of the stringed loop displaced at $\mathbf{r}$ around the same face, as shown in Fig. 2.

It is possible to see, using Eq. (4) with $\lambda = c_{ij} \in C$, that $c_{ij}^{(g_\mathbf{r})} = l_{\mathbf{r},c_{ij}}$. By means of Proposition 1.1, the r.h.s. of this equation can be decomposed as the discrete Wilson-loop around face 'ij' at $\mathbf{r}$, with the strings connecting $e$ to $g_\mathbf{r}$ *omitted* – hereafter, we denote by Wilson-loop the untraced ordered product of link around a closed line. Indeed, from Eq. (E.7) one readily verifies that those links vanish.

We conclude that the commutator $c_{ij}^{(g_\mathbf{r})}$ can be directly interpreted with the C-valued Wilson-loop around face (ij) at $\mathbf{r}$ without strings attached.

# 4  CS-crystals as Gauge Translation Groups

In this section, we shall uncover how any Cayley-crystal can be seen to host the one-body dynamics associated to a GTG. A CS-lattice equipped with a translation-invariant Hamiltonian will be referred to as a CS-crystal.

## 4.1  Schrödinger equation on the $C$-invariant subspaces

Recall the definition of induced representations (see Appendix C.1). Any regular representation of a group can be induced by the regular representation of a subgroup. In particular, with $C$ such subgroup, it holds

$$\mathcal{T}^L = \mathrm{Ind}_C^G(\mathcal{T}_C^L) \overset{def}{=} \bigoplus_\xi \mathcal{T}_\xi^{\oplus d_\xi} \tag{6}$$

where $\mathcal{T}_C^L$ is the left regular representation of $C$, which we decomposed into a sum of representations $\mathcal{T}_\xi = \mathrm{Ind}_C^G(\sigma_\xi)$ of $G$; $\sigma_\xi$ is a $d_\xi$-dimensional irrep of $C$ labeled by $\xi$, which appears $d_\xi$ times, according to Peter-Weyl theorem, up to unitary equivalence.

Given a basis $\{e_\xi^{(\alpha)} \in V_\xi \,|\, \alpha = 1,\ldots,d_\xi\}$ of the vector space $V_\xi$ on which $\sigma_\xi$ acts, the quantities

$$\sigma_{\xi,c}^{(\mu\nu)} = \langle e_\xi^{(\mu)}, \sigma_{\xi,c}(e_\xi^{(\nu)})\rangle \tag{7}$$

are the matrix coefficients of the unitary representation $\sigma_\xi$. Then, by Peter-Weyl theorem, the $C$-valued functions $f_{\xi\mu\nu}(c) = \{\sqrt{d_\xi}\,\sigma_{\xi,c}^{(\mu\nu)}\}_{\mu\nu}$ form an orthonormal basis of the $\xi$-invariant space in $L^2(C)$. In particular, the spans of the subsets $B_{\xi\mu} = \{f_{\xi\mu\nu}\}_\nu$ equal $d_\xi$-dimensional invariant subspaces that are orthogonal to each other. [42].

Let us denote with $\vec{\mathcal{B}}_{\xi\mu}(\mathbf{r})$ a generic function in the invariant subspace associated to the $\mu$-th copy of $\mathcal{T}_\xi$, here the vector elements are expressed in the basis $B_{\xi\mu}$. Restricting to one constituent in the decomposition of Eq. (6) and applying the definition of induced representations (given in Appendix C.1) we get the following Schrödinger equation,

$$i\frac{\partial}{\partial t}\vec{\mathcal{B}}_{\xi\mu}(\mathbf{r}) = \sum_\lambda \kappa_\lambda\,\sigma_{\xi,l_{\mathbf{r}-\mathbf{r}_\lambda,\lambda}} \cdot \vec{\mathcal{B}}_{\xi\mu}(\mathbf{r}-\mathbf{r}_\lambda)$$

$$\overset{def}{=} \sum_\lambda \kappa_\lambda\,\mathcal{T}_{\xi\mu,\mathbf{r}_\lambda}^\pi(\vec{\mathcal{B}}_{\xi\mu})(\mathbf{r}), \tag{8}$$

where $\xi$ labels the $C$-irreps equivalence classes, $\mu = 1,\ldots,d_\xi$, $\sigma_{\xi,\lambda}$ is the unitary matrix of Eq. (7) and we defined the maps $\mathcal{T}_{\xi\mu}^\pi$, which will be discussed in the next paragraph.

The above equation states that every hopping is accompanied by a unitary rotation and is completely analogous to that of a particle bearing a static gauge field. Therefore, a generic

wavefunction that has amplitude over multiple irreps subspaces will behave as a particle having multiple charges, each of which bears a Wilson-loop configuration $\{\sigma_{\xi, c_{ij}^{g_r}}\}_{i,j,\mathbf{r}}$. Notice that charges associated to copies of the same representation bear the same configuration.

## 4.2  Generalized projective representations

Each of the Hamiltonians in Eq. (8) commutes with *gauge* translation operators $\mathcal{T}_{\xi\mu}^{\pi} : \mathbb{Z}^n \to \mathcal{U}_{\xi\mu}$, where $\mathcal{U}_{\xi\mu}$ denotes the unitaries on the functions $\vec{\mathcal{B}}_{\xi\mu}(\mathbf{r})$. Such operators satisfy (we suppress the irreps labels)

$$\mathcal{T}_{\mathbf{r}_1}^{\pi} \mathcal{T}_{\mathbf{r}_2}^{\pi} = \Sigma(\mathbf{r}_1, \mathbf{r}_2)\, \mathcal{T}_{\mathbf{r}_1 + \mathbf{r}_2}^{\pi}, \tag{9}$$

with

$$\Sigma(\mathbf{r}_1, \mathbf{r}_2) = \oplus_{\mathbf{r}}\, \sigma_{\xi,\, l_{-\mathbf{r}_1 - \mathbf{r}_2, g_{\mathbf{r}_2}}^{(g_{\mathbf{r}})}},$$

which is a generalized version of a *projective* representation with $d_\xi \geqslant 1$.

We stress that these representations admit a lift to the linear representation $\mathcal{T}^L$ isomorphic to $G$ [43]. Therefore, $G$ can be seen as a *gauge* translation group related to each of the Hamiltonians in Eq. (8). We remark that, given a Wilson-loop configuration, the GTG might not be unique, the obvious example given by the trivial irrep of $C$ for which all integrals of $C$ are GTGs. Importantly, the GTG is realized in a CS-crystal without any complex hopping term, that is, with real $\kappa_\lambda$s. The complexity of the hoppings emerges strictly in the dynamics subspaces. Moreover, complex $\kappa_\lambda$s would not modify the physics in any way. They imply non-vanishing abelian Peierls phases attached to the cables but, since all fluxes are vanishing, they can be gauged away.

One-dimensional representations lead to discrete-flux magnetic Hamiltonian where hoppings are accompanied by phases. In particular, the trivial representation describes a trivially gauge-less $nD$ hypercubic crystal, and the corresponding invariant space is made up of wavefunctions that are constant within each pillar. Such states have been dubbed *abelian* in the recent literature [21, 24], since they are not affected by the non-commutativity of translations group and, as such, behave as if they were on a Euclidean crystal. By contrast, all other representations host *nonabelian* states that, upon left translations, get mixed among each other by a unitary matrix within the irrep-invariant subspace (cf. more details are found in Appendix C.3).

# 5  Construction of GTGs from a desired gauge group

## 5.1  Groups integrability

While any group has a commutator subgroup, the converse is not true. Since in the math literature [44] the commutator group is also named the *derived* group, this issue concerns the existence of *integrals* of a given group $C$, that is, the existence of a group whose derived group is $C$. Rephrased in the graphical context, the pillar of a nontrivial lattice (that is, comprising more than just one pillar) cannot be made to be any group $C$. Rephrased in the physical context, not all gauge groups admit a GTG. Implicit necessary and sufficient conditions for integrability have yet to be found, but the issue is considered hard to solve, involving non-abelian group cohomology theory [45]. Nevertheless, the integrability or non-integrability of many classes of groups have been assessed in recent years, and the reader is referred to these works [46–49] for a much richer discussion on the topic. For instance, abelian and alternating groups are integrable while almost all symmetric groups are not. The smallest non-integrable group is thus $S_3$, the symmetric group of order 6. Among the integrable ones, the obvious

integrable groups are the perfect ones, defined to be their own derivative, analogs of the exponential functions. *Explicit* necessary and sufficient conditions for integrability, requiring specific group presentations, have recently been found [38, 50]. In particular, the author of Ref. [38] was able to work out explicit presentations of a *universal* class of integrals that are defined as *n*-dimensional *maximal integrals*. They have the property that their abelian support is $T_n^{ab}$ for some $n$. Importantly, *all* other integrals of $C$, possibly with $\tilde{n} \neq 0$, can be obtained from them by a quotienting procedure (see Proposition 3.1 adapted from [38] and Section 7 for the procedure).

## 5.2 Integrals of cyclic groups

In Section 9 we shall discuss the maximal integrals of the smallest cyclic groups. We will not treat non-abelian cases since they require the introduction many technical details. Some non-abelian cases are discussed, without physical implications, in Ref. [38].

A theorem from the same reference, restricted to the case of $C$ cyclic, claims that *all n*-dimensional maximal integrals of $C$ can be presented as

$$
\begin{aligned}
G &= \langle \{1, 2, \ldots, n\} \,|\, \{c_{12}^m\} \cup R_p \cup R_a \rangle \\
R_p &= \{\, \bar{c}_{ij} \, c_{12}^{p_{ij}} : 1 \leqslant i < j \leqslant n \,\} \\
R_a &= \{\, \bar{c}_{12}^{(k)} \, c_{12}^{a_k} : 1 \leqslant k \leqslant n\} \qquad (n \geqslant 2),
\end{aligned}
\tag{10}
$$

where $p_{ij} \in \mathbb{Z}_m$ and $a_k \in \mathbb{Z}_m^\times$ – here $\mathbb{Z}_m^\times$ is the multiplicative group of integers modulo $m$, which is not isomorphic to $\mathbb{Z}_m$ unless $m$ is a prime number greater than 2 – satisfy

$$
p_{ij}(a_k - 1) = p_{kj}(a_i - 1) - p_{ki}(a_j - 1) \quad (\text{mod } m)
\tag{11}
$$

for $1 \leqslant k < i < j \leqslant n$, $p_{12} = 1$.

In these presentations, the way $C$ embeds in $G$ can be understood as follows. The cyclic group is generated by a single element, embedded as $c_{12}$. The first relation implies the cyclic property. $R_p$ implies that all other parent commutators are a specific multiple of that generator, see Fig. 2(left). $R_a$ imposes that the nearest-adjacent commutators $c_{12}^{(k)}$ are also some multiple of the generator, see Fig. 2(right). No further constraints are needed, e.g. on the second-adjacent commutators, since the required group structure of $G$ already fixes them.

The geometric content of the presentation Eq. (10) is quite clear. The first relator sets the size of the pillar or, seen as a GTG, it imposes that in each irrep sector the magnetic fluxes through the face '12' on the abelian support is a power of the flux $\Phi_0/m$ with $\Phi_0$ the flux quantum. The other sets $R_{p/a}$ act on the disposable degrees of freedom near the central hypercube by fixing their fluxes as a power of that of face '12' at the origin.

We observe that if $n = 2$ then $R_p$ is trivial, Eq. (11) is always satisfied and many properties of the integrals can be easily characterized (see Appendix E.4).

# 6 Unit cells of CS-lattices

The analysis of the subgroups of $G$ is crucial in order to implement PBCs in CS-lattices (see the following paragraphs) and to perform a more advanced analysis of the dynamics of Eq. (8) (see Section 8.1).

Any subgroup of $G$ defines a lattice tesselation via a coset-decomposition similar to Eq. (3). However, tiles defined by a generic subgroup lack any specific property. Here we show that there are always two subgroups, $P$ and $\tilde{Z}$ defined below, which deserve special attention. The definitions and properties of all cells introduced here are summarized in Table 1.

## 6.1  Standard periodicity

CS-lattices defined from a maximal integral of a finite group $C$ happen to be periodic. Thus, there are always $n$ linear independent vectors $\mathbf{m} \in \mathbb{Z}^n$, such that for all $\lambda \in \Lambda$ and $\mathbf{r} \in \mathbb{Z}^n$, $l_{\mathbf{r},\lambda} = l_{\mathbf{r}+\mathbf{m},\lambda}$, forming a supercell whose pattern of cables tessellates the lattice. We refer to these supercells as *standard CS-cells* and to those that are minimal (not containing other ones) as *unit cells*. A sketchy example of a standard unit CS-cell can be found in Fig. 1b, represented as the blue area inside the lattice. The cell is meant to comprise all the pillars and cables within the area and the *outgoing* ones that, since the Cayley-graph is directed, can be fixed to be the ones starting inside and having a dangling end either to the right or to the top. We prove in Theorem 1.5 a necessary and sufficient condition for $\mathbf{m}$ to set the periodicity. Such vectors are found to be independent from the set $\Lambda$ and determine an abelian subgroup $P$ whose elements, acting by left multiplication on $G$, shift the lattice by a finite number of CS-cells. In the same theorem we show also that $P$ is nontrivial. This feature is not *a priori* obvious since there could have been instances of aperiodic, e.g. quasicrystals-like, maximal integrals. If $n = 2$ we show in Corollary E.4 that standard CS-cells are necessarily rectangular (without any tilt). This feature may not be true in general when $n > 2$, see the remark after the corollary.

   We notice that, since $P$ has a finite index in $G$ (i.e., finite transversal), such maximal integrals and thus the GTGs we consider here are virtually abelian groups, that is, groups containing an abelian subgroup of finite index.

## 6.2  Hidden periodicity

In Appendix B.1 we show that the center group (made of elements that commute with all elements of $G$) can be factorized as $\widetilde{Z}Z_{CG}$ where $Z_{CG} = Z \cap C$ and $\widetilde{Z} \cap Z_{CG}$ is trivial. $\widetilde{Z}$ is abelian, isomorphic to $\mathbb{Z}^n$ but is not uniquely defined if $Z_{CG}$ is nontrivial (cf. Section 9 for instances of non-uniqueness), therefore we shall add a label to this group. Therefore, we can use $Z$ to tessellate the lattice and define what we refer to as the *primitive* unit CS-cell. The groups $\widetilde{Z}_i$ shift the cell along the abelian support whereas $Z_{CG}$ (again abelian) shifts vertically along the pillars (we expand on this point in Appendix B.2). It is clear that primitive CS-cells do not comprise entire pillars but only a reduced part of them if $Z_{CG}$ is nontrivial. Moreover, primitive unit CS-cells do not repeat over the lattice forming a periodic pattern, that is, for a generic lattice $G$, it is possible to find a $z \in Z$, a position $\mathbf{r}$ and a hopping $\lambda$ such that $l_{\mathbf{r}+\mathbf{r}_z,\lambda} \neq l_{\mathbf{r},\lambda}$.

   Despite these unfortunate features, the primitive CS-cell constitutes the fundamental piece of the lattice because of two key properties: the groups $G/\widetilde{Z}_i$ (which is in correspondence to a *primitive CS-supercell*) are the minimal integral of $C$, and as such it allows one to identify all possible valid periodic boundary conditions on $G$; when a Hamiltonian is given and the embedding of $Z$ in $G$ is known, it is possible to obtain a full eigenspace decomposition of the dynamics. We shall see these features, respectively, in the next paragraph and in Section 8.1.

## 7  Periodic boundary conditions

Periodic boundary conditions (PBCs) are implemented in Cayley-crystals by means of normal subgroups $N \lhd G$. The elements of $N$ identify which sites of the lattice have to be collapsed to the same point and the resulting lattice is again a group, namely $G/N$. Importantly, since we want to keep the commutator group of $G$, the additional requirement is necessary that $N \cap C$ be trivial. We shall refer to the groups $N$ with these properties as *valid PBCs*.

   Physically, the implementation of PBCs on an infinite CS-lattice corresponding to a maximal integral proceeds in two steps:

i) (*cutting step*) A convenient representative of $G/N$ in $G$ is chosen to constitute the PBC-lattice vertices. We suppose such representative to be connected, convex and made of contiguous pillars (e.g. some multiple of the blue area in the sketchy case shown in Fig. 1b). Eventually, if $N^{ab} \cong \mathbb{Z}^n$ then the abelian support of the PBC-lattice, $(G/N)^{ab}$, is a finite subset of $G^{ab}$ and isomorphic to a product of $n$ cyclic groups.

ii) (*glueing step*) The cables that are in the interior of the chosen representative are left unchanged but those across the boundaries – all pairs of ingoing and outgoing cables at opposite faces of the representative – should be pairwise *glued* to respect the new group structure. These connections may produce either cables that agree with the pattern of the pristine lattice cables or disagree, creating *twisted* connections. We show in Lemma 1.7 how the cable formula Eq. (5) may get modified by a twist factor at the boundary. We observe that the addition of boundary cables in a physical implementation breaks the nice bulk hypercubic structure of the lattice. However, since their number is small in comparison with the number of the bulk ones, it should be possible to arrange them without much effort.

## 7.1 Valid PBC

Among all valid PBCs, a class of groups stands out, which have the property that the cut-and-glue procedure does not cut across the periodic pattern of their cabling nor it creates twists at the boundaries. We name these groups *standard PBCs* and, in some way, they induce the most close generalization of the PBCs of Euclidean lattices. As shown in Proposition 2.6, they are exactly the subgroups of $Q = P \cap Z$ (we refer to the cells associated to the group $Q$ as *standard CS-supercell*). Some properties of this group are described in Proposition 2.3 and Corollary 2.5. All other PBCs need twists at the boundaries. Standard and twisted PBCs are all determined by the primitive unit CS-cell in the following way. In Proposition 3.2 we show that valid PBCs need to be central in $G$, that is, $N \leqslant \widetilde{Z}_i$ for all $i$'s. Thus, all lattices with valid PBCs must be tessellated by the primitive CS-supercells defined by the transversal of some $\widetilde{Z}_i$, as mentioned in Section 6.2. The shape of such supercell is discussed in Appendix C. We notice that the twisted PBCs related to the $i$-th CS-supercell type are in one-to-one connection with the elements of $(\widetilde{Z}_i \backslash Q)$.

## 7.2 Non-Cayley PBC

The group $P$ defined in Section 6.1 is not necessarily normal in $G$. In such cases, the transversal of $X$ in $G$, with $X \leqslant P$, still identifies a periodic lattice through the same cut-and-glue procedure described above (with no twist at the boundaries), but it will not correspond itself to a group. As a consequence the resulting lattice is not a Cayley one neither it can be a GTG. We show in an example case in Appendix D.3.1 the impact of such PBCs on the spectral property of the Hamiltonia. The set of allowed quantum numbers loses its homomorphic correspondence with the irreps of the translation operators, exactly as it happens in magnetic crystals threaded by a fractional flux quantum [51], where all momenta get a finite shift.

# 8 Bloch decompositions of CS-crystals

In this section we shall discuss several *Bloch* decompositions of the Hamiltonian in Eq. (2) that are more refined than that in Section 4.1.

| Crystal | Cell type | Tessel. group | Periodic | PBC |
|---|---|---|---|---|
| Euclidean | standard | $\mathbb{Z}^n$ | ✓ | valid + standard |
| Cayley-Schreier | standard (unit) | $P$ | ✓ | non-Cayley |
| | standard (supercell) | $Q$ | ✓ | valid + standard |
| | primitive (unit) | $Z$ | ✗ | non-valid |
| | primitive (supercells) | $\widetilde{Z}_i$ | ✗ | valid |

Table 1: Relevant CS-cells and their properties. The Euclidean lattice case is considered on top as a reference. (first column, after the double vertical line) Defining *tessellation* group. (second column) Has the cell pattern a Euclidean-like periodicity? (last column) Generic attributes of the PBCs lattice.

## 8.1 Full and partial Bloch decompositions

The Hamiltonian in Eq. (2) can be fully block decomposed as [24, 52]

$$
\begin{aligned}
\mathcal{H} &= \bigoplus_{d>0,\xi_d} \mathcal{H}_{\xi_d}^{\oplus d}, \\
\mathcal{H}_{\xi_d} &= \sum_{\lambda \in \Lambda} \kappa_\lambda \mathcal{T}_{\xi_d,\lambda}
\end{aligned}
\tag{12}
$$

where $\xi_d$ labels the $d$-dimensional $G$-irreps equivalence classes; since $\mathcal{T}_{\xi_d}^L$ is a regular irrep, each block $\mathcal{H}_{\xi_d}$ appears $d$ times in the direct sum [53].

Unfortunately, the full group structure of $G$ is not known *a priori* when it is built as an integral of $C$ (see Section 5). As a consequence, the above decomposition cannot be carried out. However, the knowledge of the groups associated to the periodic cells (see Table 1) allows us to make further progress from Eq. (8) and obtain a partial Bloch decomposition.

Let us proceed as with Eq. (6) but with the subgroup given by $YC$ with $Y$ a central subgroup with trivial intersection with $C$, such as $Q$ or any $\widetilde{Z}_i$ (notice $\widetilde{Z}_i C$ does not depend on $i$). The Hamiltonian is decomposed accordingly as (see Appendix C.2 for the derivation)

$$
\begin{aligned}
\mathcal{H} &= \bigoplus_{\eta,\xi} \sum_{\lambda \in \Lambda} \kappa_\lambda \, \chi_{\eta,\lambda}^G \, \widetilde{\mathcal{T}}_{\eta\xi,\lambda}^{\oplus d_\xi}, \\
\widetilde{\mathcal{T}}_{\eta\xi} &= \bar{\chi}_\eta^G \, \mathrm{Ind}_{YC}^G \left( \chi_\eta^Y \sigma_\xi \right),
\end{aligned}
\tag{13}
$$

where $\chi_\eta^Y$ is a (1D) representation of $Y$ whereas $\chi_\eta^G$ is a multiplicative representation of $G$ into $L^2(G)$ acting as $\chi_{\eta,g}^G = e^{-i\eta \cdot \mathbf{r}_g}$ and coinciding with $\chi_\eta^Y$ on $Y$; $\eta$ parametrizes the $n$-dimensional toric $BZ$ defined by the group $Y$ (we shall denote it by $pBZ$ if $Y = \widetilde{Z}_i$); each $\widetilde{\mathcal{T}}_{\eta\xi}^L$ acts on a Hilbert space of finite dimension $d_\xi |G/(YC)|$ with $\xi$ labelling the irreps $\sigma_\xi$ of $C$ as in Section 4.

This decomposition highlights its *Bloch* character. In fact, the plane-wave sector, which describe the transport among cells, is made explicit and is controlled by the irreps $\eta$ of $Y$. We observe that such a decomposition is saturated when $Y = \widetilde{Z}_i$.

The span of Bloch wavefunctions $\{\mathcal{B}_{\eta\xi\mu}^{(n\nu)} : 1 \leqslant n \leqslant |G/(YC)|, 1 \leqslant \nu \leqslant d_\xi\}$ equals the space on which $\widetilde{\mathcal{T}}_{\eta\xi}$ acts. Their specific expression, together with a Bloch orthonormal basis, is given in Appendix C.3. The Bloch *Hamiltonian eigenstates* can be found by applying a unitary rotation

on the Bloch basis. If $\sigma_\xi = 1$ is the trivial representation, then $\widetilde{\mathcal{T}}_{\eta\xi}$ is an abelian representation and the Bloch Hamiltonian eigenstates are also eigenstates of the translation operators. For all other representations, non-trivial 1D representations included, it can be checked again that the Bloch eigenstates (at different energies but same invariant space) get mixed among each other upon left translations, whereas they mix within the degenerate multiplet (at different $\mu$s) upon right translations since the Hamiltonian commutes with the right regular representation.

## 8.2  Other Bloch decompositions

Finally, we can consider the regular representation induced from the product $PC$ (see Appendix C). Since $P$ is not necessarily normal, degeneracies are not well highlighted by the associated Hamiltonian decomposition. However, in general, there is no inclusion relation among $P$ and $\widetilde{Z}$ and the two decompositions may display different invariant subspaces. Thus, the intersections of these invariant subspaces are again invariant and define a more refined decomposition that may turn useful in specific analysis. Another group that can be considered is the centralizer of $C$ in $G$, since $C_C(G) \geqslant Z$. We show in Proposition 2.7 that, for $n = 2$ and $C$ cyclic, its presentation is known so long as the one of $G$ is given.

In the simple examples that we shall show in the next section, the partial Bloch decompositions along $\widetilde{Z}C$ allow to retrieve all irreducible subspaces and the full decomposition. However, for more complex examples this nice feature might not be lost. The bridge between the partial decomposition and the full one in Section 8.1 is found within the so-called MacKey theory [54, 55] that, however, requires further knowledge of the subgroup structure of $G/\widetilde{Z}$. However, in practical realizations the order of $G/\widetilde{Z}$ is not going to be large (in a group-theoretic sense) and its irreps can be read out from the GAP library or other databases [56].

# 9  Examples of CS-crystals

In this work, we restrict the examples to cyclic commutators, the smallest non-trivial groups being $C = \mathbb{Z}_2, \mathbb{Z}_3$. The simplest abelian support to treat is a two-dimensional one. The CS-lattices are built from the canonical presentations in Section 5.2.

The group $C$ has only one generator $c$ that must be embedded in $G$ necessarily as the terms $c_{12}$ (also shorten to $c$). The irreps of $C = \mathbb{Z}_m$ will be denoted by $\sigma_\omega$ with $\omega \in \{e^{2\pi i k/m} : k = 1, \ldots, m\}$. In Eq. (10) the set $R_p$ is trivial and the only degree of choice will come from the set $\mathcal{A} = \{a_1, a_2\}$ chosen to determine $R_a$. Due to Theorem 1.5 and Propositions 2.7, 4.1 and 4.2 all subgroups of interest can be computed explicitly without effort.

In the next sections, we shall investigate only the CS-lattices and their GTGs, confining the analysis of the spectra of the Hamiltonian to Appendix D. For simplicity of the treatment, we will consider the Hamiltonian Eq. (2) with NN couplings only, i.e. $\lambda = 1, 2$.

## 9.1  $C = \mathbb{Z}_2$ and $n = 2$

Since $\mathbb{Z}_2^\times \cong \mathbb{Z}_1$, necessarily $\mathcal{A} = \{1, 1\}$ and $R_a = \{\bar{c}^{(1)}c, \bar{c}^{(2)}c\}$. The presentation of the unique maximal integral is given by

$$H = \langle 1, 2 \,|\, c^2, \bar{c}^{(1)}c, \bar{c}^{(2)}c \rangle \tag{14}$$

The CS-lattice is shown in Fig. 3. It is quite evident from the pattern in the lattices that $P = \langle 1^2, 2 \rangle_{ab}$ (the 'ab' subscript here denotes abelian generation), which is in agreement with the expectation, by Corollary E.4, that $P_{\text{straight}} = P$. Moreover $Q = \langle 1^2, 2^2 \rangle$ and $C = Z_{CG}$ is central while $C_G(C) = G$. The group $\widetilde{Z}$ is not unique since $Z_{CG}$ is nontrivial (cf. Appendix B.1). All $\widetilde{Z}_i$s are generated by the generators of $Q$, multiplied by either $e$ of $c$, cf. Proposition 4.2. If

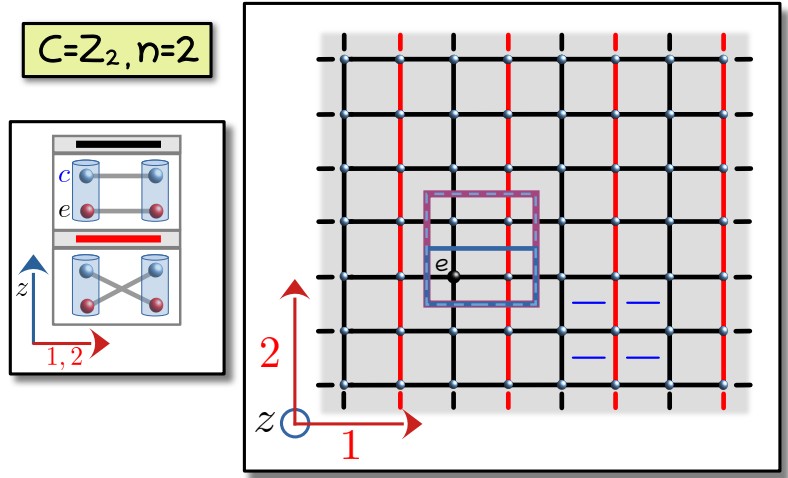

Figure 3: Two dimensional Cayley-Schreier lattices with $C = \mathbb{Z}_2$. The cables listing is on the left panel. The main cells are shown on the abelian support: standard unit CS-cell (blue), standard CS-supercell (light blue, dashed), primitive unit CS-cell (purple). The flux pattern in the trivial and non-trivial irreps of $C$ can be inferred from the loops values shown in blue at one side of the lattice (we mapped $(e, c) \rightarrow (e, -)$). In both cases it is homogeneous.

$\widetilde{Z} = Q$, then the associated primitive CS-supercell group is a dihedral one, $G/\widetilde{Z} = D_8$. In the other three cases one finds it to be the quaternionic group $Q_8$.

## 9.2    $C = \mathbb{Z}_3$ **and** $n = 2$

Since $\mathbb{Z}_3^{\times} \cong \mathbb{Z}_2$, we have four possible choices for $\mathcal{A}$. However, only three of them lead to non-isomorphic GTGs. In Table 2 we list the details about the four CS-lattices. Case 'a' is the central extension case; cases 'b' and 'c' are equivalent up to the exchange of letter $1 \leftrightarrow 2$ in the maximal integral presentation; case 'd' is arguably the most complex case.

    The CS-lattices relative to the four cases are shown in Fig. 4. Several remarks are in order. First, there is a reduction of the standard CS-supercell size the along the direction $k$ in which the corresponding $a_k$ is non-trivial. This is an expected consequence of Proposition 4.1, by which the size is $m$ if $a_k = 1$ trivial but can be less otherwise. Second, as mentioned, the cases 'b' and 'c' are isomorphic, however, this does not emerge clearly looking at the pattern of cables. Nevertheless, the sizes of standard CS-supercells and primitive unit CS-cell are the

| Case | $\mathcal{A}$ | $R_a$ | $P$ | $Q$ | $\widetilde{Z}$ | $Z_{\mathrm{CG}}$ | $\mathcal{C}_G(C)$ | $G/\widetilde{Z}$ |
|---|---|---|---|---|---|---|---|---|
| a | $\{1,1\}$ | $\{\bar{c}^{(1)}c, \bar{c}^{(2)}c\}$ | $\langle 1^3, 2 \rangle$ | $\langle 1^3, 2^3 \rangle$ | $Q$ − affine | $C$ | $G$ | $U(3,3), C_9 \rtimes C_3$ |
| b | $\{2,1\}$ | $\{c^{(1)}c, \bar{c}^{(2)}c\}$ | $\langle 1^2, 2 \rangle$ | $\langle 1^2, 2^3 \rangle$ | $\langle 1^2, 2c \rangle$ | ✗ | $C\widetilde{Z}$ | $S_3$ |
| c | $\{1,2\}$ | $\{\bar{c}^{(1)}c, c^{(2)}c\}$ | $\langle 1^3, 2^2 \rangle$ | $P$ | $\langle 1\bar{c}, 2^2 \rangle$ | ✗ | $C\widetilde{Z}$ | $S_3$ |
| d | $\{2,2\}$ | $\{c^{(1)}c, c^{(2)}c\}$ | $\langle 1^2, 2^2 \rangle$ | $P$ | $\langle 12c, \bar{1}2c \rangle$ | ✗ | $C\widetilde{Z}$ | $S_3$ |

Table 2: Details about the four CS-lattices with $C = \mathbb{Z}_3$ and $n = 2$. In case 'a', $\widetilde{Z}$ can be defined in several "affine" ways from the generators of $Q$ (cf. similar case in Section 9.1), which leads to 2 possible primitive CS-supercell groups (last row). The cross symbol denotes the trivial group.

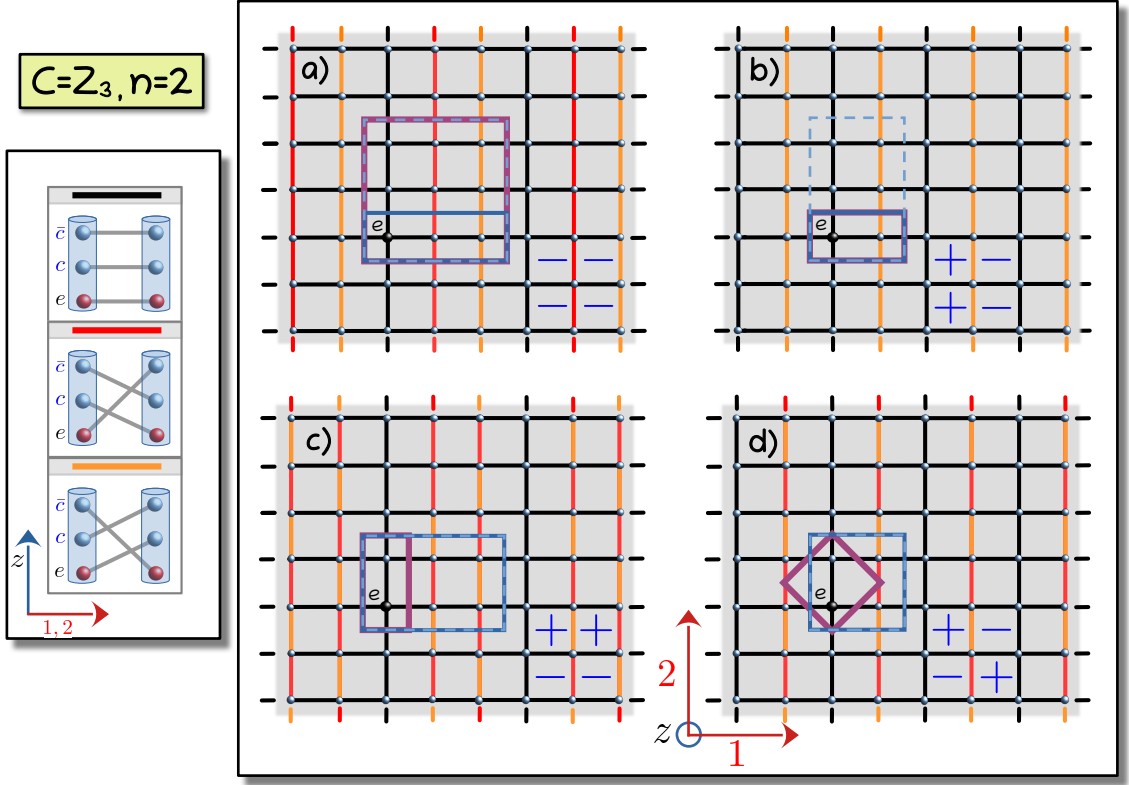

Figure 4: Same as in Fig. 3 with $C = \mathbb{Z}_3$. In this case 4 possible CS lattices exist.
. The flux pattern in all 3 irreps of $C$ can be inferred from the loops values shown at one side of the lattice (we mapped $(e, c, \bar{c}) \to (e, -, +)$).

same (up to a rotation). This is expected since such cells are defined by central groups, defined by their implicit commutation property, while $P$-groups have a graph-dependent definitions. The flux patterns also coincide up to rotations. Third, 'c' and 'd' are instances where the primitive CS-cell truly identifies a hidden periodicity. Four, one can check how the claims in Corollaries 4.3 and 4.4 apply to these cases in a different way. In particular, in case 'a' $P > \tilde{Z}$ since the lattice generators commute with $C$, in case 'b' the same holds even if the commutation fails while the inclusion fails in cases 'c' and 'd'. Finally, we observe, looking at the group $S_3$ in the last column in the table, that the primitive CS-supercell groups may coincide even for non-isomorphic CS-lattices. A theorem in Ref. [38] shows that non-isomorphic Cayley-lattices always lead to non-isomorphic lattices after imposing PBCs if the generators of $\tilde{Z}$ are not in the PBCs group. Moreover, we observe that, since $S_3$ is the smallest non-abelian group, $S_3$ is also the smallest GTG of all possible gauge groups.

## 10 Discussion

### 10.1 Wilson-loop configurations

It is natural to ask what kind of gauge configurations appear in Eq. (8).

Local shuffling of the pillars elements determines a redefinition of links as $l'_{\mathbf{r},\lambda} = P_{\mathbf{r}+\mathbf{r}_\lambda} l_{\mathbf{r},\lambda} P_{\mathbf{r}}^T$, with $P_{\mathbf{r}}$ some permutation matrices. Since such reassembling should not change the physics it is convenient to ask the question not in terms of the set of hopping matrices configurations $\{\sigma_{\xi, l_{\mathbf{r},\lambda}} : \mathbf{r} \in \mathbb{Z}^n, \lambda \in \Lambda\}$ but rather in terms of the set of Wilson-loop configurations

$\{\sigma_{\xi, c_{ij}^{(g_\mathbf{r})}} : g_\mathbf{r} \in T_n^{ab}, 1 \leqslant i < j \leqslant n\}$, which are invariant under such permutations (assuming without loss of generality $P_{\mathbf{r}=\mathbf{0}} = \mathbb{1}$). Since these configurations depend on the pillar elements $c_{ij}^{(g_\mathbf{r})}$, finally the question is about the types of possible embeddings of $C$ in a larger group, the issue addressed in Sections 5.1 and 5.2.

From the example in Section 9.1 we saw that there is only one possible flux configuration if $C = \mathbb{Z}_2$. In particular, all loop fluxes around the 4-plaquettes are $\pi$ and, for instance, no vanishing flux appears. Moreover, Corollary 2.5 proves that the Wilson-loops around the standard CS-supercell in any irrep sector of $C$ vanish. Thus, it seems that CS-crystals admit only few Wilson-loop configurations. However, it can be shown that it is possible to construct *any* (effectively-)$\mathbb{Z}_m$ flux configuration having $p$ freely-chosen fluxes in a periodic cell, in a CS-crystal with $C = (\mathbb{Z}_m)^p$ and $n = 2$. At present, we do not know whether this is possible also in the non-abelian case. However, since such topic goes beyond the scope of this paper, it will be treated separately somewhere else. In any case, even in the case that CS-crystals may realize any configuration, we must mention that they are redundant systems if one is only interested in reproducing the dynamics of particles under a *single* taylored Wilson-loop configuration. Indeed, either one can opt for a standard implementation, using unitary matrices and local Hilbert spaces whose dimension equals that of the unitaries, or, if real hopping amplitudes are required, it is easy to envision systems where the pillars size need not be enhanced by any power. For instance, with 2D abelian supports, if links can be adjusted at will, there are enough degrees of freedom to impose any $\mathbb{Z}_m$ Wilson-loop configuration in a lattice with pillars of size $m$. Therefore, CS-crystals are interesting primarily because they represent a whole group, $G$, and not just some (generalized) projective representation (cf. Eq. (9)).

Finally, we can ask what is the periodicity of the Wilson-loop configuration in a given CS-crystal with group $G$. Assuming $C$ cyclic it is possible to easily answer such a question. The minimal *common* periodicity, i.e. common to the flux configurations of all irreps subspaces, is given by the group $\mathcal{C}_G(C)/C$, with $\mathcal{C}_G(C)$ the centralizer of $C$ in $G$ as described in Proposition 2.8. As a corollary, since $\mathcal{C}_G(C) \geqslant P, \widetilde{Z}$, it follows that the *universal* flux-periodic cell, defined by this minimal periodicity, is smaller than or equal to both the primitive and the standard unit CS-cells. We do not know the answer in the non-abelian case.

## 10.2 Comparison with existing literature

Our proposed method for constructing GTGs and the associated CS-crystals from a desired gauge group is based on the short exact sequence

$$0 \to C \to G \to \mathbb{Z}^n \to 0. \tag{15}$$

In the literature [50, 57, 58], for a number of different purposes, it is often considered a "reversed" method based on central extensions of the form

$$0 \to \mathbb{Z}^n \to G \to H \to 0. \tag{16}$$

The second method applies in our case, recognizing that the central group $\mathbb{Z}^n$ is $\widetilde{Z}$ while $H$ is $G/\widetilde{Z}$, the primitive CS-supercell group. We notice that our perspective is quite different, as in the latter $H$ (the primitive CS-supercell group) is fixed, while in the former its commutator group is. The second approach is preferable if the concern is in shaping the dynamics, as the irreps of $\widetilde{Z}$ refine the invariant subspaces better than those of $C$. However, since $C$ will always be contained in the primitive CS-supercell group, constructing lattices from $C$ addresses more fundamental properties and, arguably, the most fundamental ones, since $C$ is the minimal group determining non-commutativity.

We notice that if $Z_{CG}$ is trivial then, in Eq. (16), $\widetilde{Z}$ embeds as a maximal abelian normal group of finite index, making $G$ an abstract crystallographic group, according to Theorem 2.2

of Ref. [58]. Crystallographic groups are extended versions of the Euclidean symmorphic space groups and partial classifications have been obtained [58].

In Section 9, we found that two of the maximal integrals constructed with Eq. (10) are isomorphic to each other. There seems to be no systematic ways to discard redundancies other than direct checks. The MacLane method, based on a central extensions version of Eq. (15), proposed to find MTGs in Refs. [10, 57], has the same issue, as noted in Ref. [14]. Concerning this approach, which is restricted to $C$ cyclic, we observe that it is less general than our method. Indeed, it allows to describe only homogeneous flux configurations and corresponds to setting $a_i = 1$ for all $i$'s. Moreover, nothing in their theory enforces the central (gauge) group to embed in $G$ as its commutator group whence, we believe, the built extension $G$ might not actually be always the desired MTG.

Finally, we remark that different dynamics on Cayley-graphs have been considered [59,60], for instance quantum walks [61] and quantum cellular automata [62]. In particular, the so-called coinless quantum walks [63, 64] on virtually abelian groups (that is, containing an abelian subgroup of finite index) closely resemble the dynamics we explored [50,65]. In those models, the coin space, which serves as an analogue of the pillar space, arises from the graph connectivity and not from an intrinsic degree of freedom of the vertices. There, it has also been shown that the dynamics can be decomposed along the nominal abelian subgroup, which in our language can be either $Q$ or $Z$, and simplified accordingly. Despite some similarities, such as the shared interest in building group extensions [50], the framework and the goals are different. The stroboscopic unitary evolution of quantum walks (or cellular automata) are much more constrained [64,66] than the Schrödinger dynamics; moreover, gauge configurations and their charges are of no concern, being a hidden structure within the coin space.

## 11 Conclusions

In conclusion, we showed that any discrete group $G$ can be regarded as the *Gauge* Translation Group of the multiple one-body dynamics appearing as Bloch constituents in its associated Cayley-crystal. This link between group theory and synthetic crystals allowed us to both generalize the notion of Magnetic Translation Groups and to provide an experimentally implementable setup for the realization of new types of dynamics. In particular, we introduced Cayley-Schreier-crystals as a scalable and universal implementation of virtually abelian GTGs. In these crystals the gauge degrees of freedom are made local in space and the bulk lattice becomes hypercubic. No complex hoppings terms are required to build such systems; a property which, combined with the fact that a neat embedding in 3D exists, makes them appealing for experiments.

Although any group $G$ can be regarded as a GTG, the converse is not true since there are groups $C$, the smallest one being $S_3$, that cannot be the gauge groups of any GTG. In this respect it would be interesting to understand the physical implications of such failure.

It is intriguing to explore more complex Hamiltonians over these crystals to exploit their peculiar group structure, in particular mixing the different irreps dynamics. Adding interactions [67,68] and disorder [69] allows one to achieve this goal. Moreover, provided complex hoppings are enabled in general, the structure of CS-lattices allows an easy implementation of Hamiltonian terms that are taylored to act differently on the specific representations. This would break the Hamiltonian symmetry while preserving the dynamics invariant subspaces. An interesting direction is to develop these systems into a full-fledged lattice gauge theory where the gauge field itself becomes dynamical. [26,70,71].

Finally, given the strong connection between group theory and the theory of higher-order topological insulators and superconductors [72], CS-crystals promise to be excellent platforms

for realizing topological systems of arbitrary nominal dimension. They also provide a natural setting to study the robust states implied by the bulk/boundary correspondence when open boundary conditions – instead of the PBCs discussed here – are implemented [73].

## Aknowledgements

The author wishes to thank Felix Flicker, Fabian Lux, Michele Burrello, Oded Zilberberg and Tomáš Bzdušek for inspiring inputs and Lavi K. Upreti for moral support and helpful discussions.

**Funding information** The work was supported by the QuantERA II Programme STAQS project that has received funding from the European Union's Horizon 2020 research and innovation programme under Grant Agreement No 101017733.

## A  2D quasi-isometric embedding of $(n > 2)$-dimensional abelian supports

The CS-lattices we have shown in the examples of Section 9 have a 2D abelian support. Equivalently, the GTGs had rank 2, that is, two generators. In general groups may have a bigger rank, posing the problem about the physically realization of these lattices in three dimensions (we assume one dimension is taken up by the pillars). An example of such groups are the hyperbolic ones, the least rank of which is 4 [23].

Here, we suggest a solution to this problem consisting in a mapping of a $n$D abelian support to a 2D one with the additional property that the length of the nearest-neighboring (NN) cables is mapped isometrically. The intuitive idea behind is shown in Fig. 5(a-b): each physical direction associated to a generator of the abelian support is pushed isometrically into the 2D plane. To ensure that group elements with different associated Schreier transversal (pillar) do not overlap, the angles among the unit vector associated to these directions must be *incommensurate*. To formalize it, we make use of the mathematical concept of field extensions [74]. For what concerns us, we are interested in field extensions of $\mathbb{Z}$; they can be written as the sets

$$\mathbb{Z}(\{r_i\}_{i=2}^n) = \{p_1 + \sum_{j=2}^n p_j r_j : p_j \in \mathbb{Z}\}, \tag{A.1}$$

which retain the specific property of their original field, i.e. closure under sum, difference, multiplication and division. An extension is fully nontrivial if the set $\{r_i\}_i$ contains reciprocally incommensurate numbers so that every specific set of numbers $\{p_j\}$ uniquely identifies an element of the extended field. For instance, $r_i = \sqrt{i^2 + 1}$ would produce an irreducible extension. Instead, $r_i = i$ would produce a completely trivial extension $\mathbb{Z}(r_1, r_2, \dots) = \mathbb{Z}$. It is customary to assume $r_1 = 1$.

In our case, since we want the abelian support to be fit in two dimensions we can think of its points as points in the complex plane. We consider Eq. (A.1) with $n$ being the number of the lattice generators and each $r_s$ being a complex number. The set $\{r_i\}_i$ must be carefully chosen so that real and imaginary parts of the set points never vanish simultaneously (at $p_j \neq 0$), otherwise we get a so-called redundant extension, where a linear dependence between the points is present.

Notice that, by choosing $r_s = e^{i\theta_s}$, the lattice will enjoy the appealing property that the links connecting NN pillars have the same length (small length adjustment are inevitably needed in

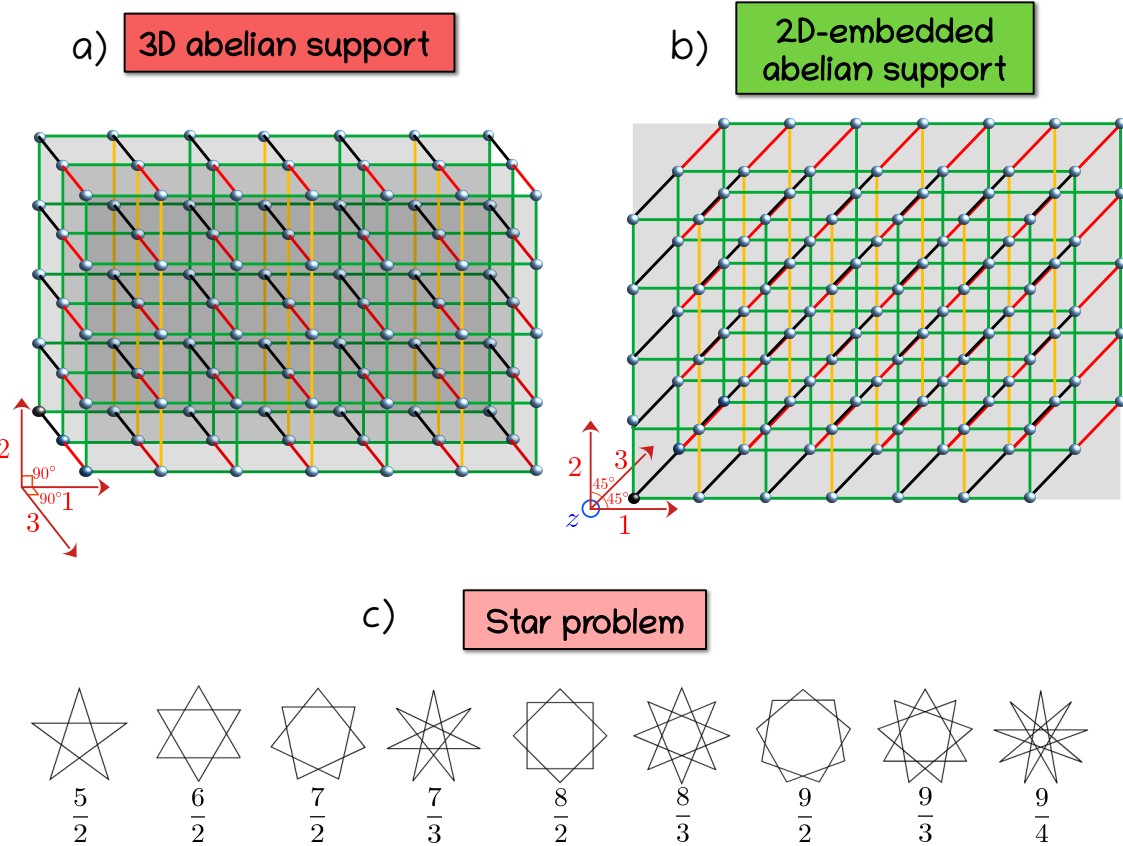

Figure 5: a) Abelian support with three generators in 3D. The coloring of cables is only for visual aid. Notice that the pillars, here collapsed to a single blue sphere, might make the connectivity very complex. b) Same CS-lattice but with the third dimensions collapsed on the plane $xy$. Pillars elements might be placed along the $z$ axis. The field extension corresponding to this arrangement has $r_2 = i$ and $r_3 = \exp\{i\pi/4\}$. c) Whenever the vectors (or their inverses) associated to the $r_i$ can be disposed to form a closed component of a star, such set determines a redundant field extension. We use the Schläfli symbols $p/q$ to denote the stars, where $p$ (the number of vertices) and $q$ (the discrete distance, on the circumcircle, between the points connected by a line) are coprime and $q \geqslant 2$.

the practice, to avoid overlaps among the links or between links and pillars). Non-NN links may get stretched or compressed, however, it can be seen that the number of cable sizes stays finite even in an infinite support. A bad choice of angles $\theta_s$ might lead to star configurations, implying that the extension is redundant, see Fig. 5(c). For example, choosing $r_2 = \exp\{-i\pi/3\}$ and $r_3 = \exp\{2i\pi/3\}$ would produce a hexagonal lattice whose points require only 2 generators to get labeled (and not 3). The hexagonal lattice has this property because the vectors associated to $\{r_i\}_{i=1,2,3}$ can be disposed to form one of the two triangles in the star $6/2$ in the figure.

Clearly, if the pillars are small enough one may also opt to use the third dimension to accommodate one more dimension of the abelian support, placing the pillar in such a way not to obstruct the links. In this case, we may define our lattice as in Eq. (A.1) but with $\{r_i\}_i$ being a set of three-dimensional *incommensurate* vectors.

# B  Details about the center group

## B.1  Decomposition and computation

In Proposition 2.2 we prove that we can decompose the center group as the direct product of subgroups

$$Z = \widetilde{Z} Z_{CG} \tag{B.1}$$

where $Z_{CG} = Z \cap C$, $\widetilde{Z} \cong \mathbb{Z}^n$ and $\widetilde{Z} \cap Z_{CG}$ is trivial. It holds,

$$Z_{CG} = \otimes_{i=1}^{\mathrm{rank}(Z_{CG})} \mathbb{Z}_{m_i} \tag{B.2}$$

where $m_i$ divides $m_{i+1}$ and by *rank* of a group we mean the minimal size a generating set can have. The latter decomposition is called *invariant factor decomposition* [75] and applies to all finite abelian groups. Differently from $Z_{CG}$, the group $\widetilde{Z}$ is not necessarily uniquely defined. Given $Z$ and $Z_{CG}$ the number of possible *complements* $\widetilde{Z}_i$ equals $|Z_{CG}|^n$, as can be seen from the fact that this number must equal the cardinality of the set $Hom(\mathbb{Z}^n, Z_{CG})$ of the homomorphisms between $\mathbb{Z}^n(\cong \widetilde{Z})$ and $Z_{CG}$, and $Hom(\mathbb{Z}^n, Z_{CG}) = (Z_{CG})^{\times n}$ [76].

The number of generators of $Z$ is $n + \mathrm{rank}(Z_{CG})$.

The explicit computation of $Z$ from a presentations of $G$ (see Section Section 5) is important if one is interested in a fine decomposition of the dynamics (see Section 8.1). The group $Z_{CG}$ is easily retrievable from the presentation of $G$ since the embedding of $C$ in $G$ is explicit in the presentations of Section 5 and one should find the elements $c \in C < G$ such that $c^{(k)} = c$. The situation is less fortunate with $\widetilde{Z}$. Being the number of its generators finite, they may be found with an extensive search among the elements of the standard CS-supercell. If $n = 2$ and $C$ is cyclic, it is possible instead to find $Z$ directly from the knowledge of the presentation parameters $a_i$ (see Proposition 4.2). The same holds for the group $Q$ (see Proposition 4.1).

## B.2  Tessellation by the primitive unit CS-cell

The primitive unit CS-cell does not stand out like the standard CS-supercell since it arises from a pattern in the links that is hidden. The former cell can be represented as subset of the latter cell since $Z \geqslant Q$. Differently from $Q$, the elements of $Z$ can have a pillar component, according to the decomposition Eq. (3). In particular, if $Z_{CG}$ is non-trivial, only a subset of the pillars elements can be retained in the primitive CS-cell definition. Instead, such pillar reduction can be avoided if $Z_{CG}$ is trivial. Indeed, the non-trivial elements of any $\widetilde{Z}_i$ are of the form $z = g_z^{ab} c_z$ with $c \notin (Z_{CG} \backslash \{e\})$. In choosing the transversal of $\widetilde{Z}_i$ in $G$ that determines a CS-supercell, which is in one-to-one correspondence with the group $G/\widetilde{Z}_i$, one may use the relation $g_z^{ab} c_z$ to remove from the cell group either $g_z^{ab}$ or $c_z$. Clearly, the *first choice* will not modify pillars and impacts only the abelian support of the cell. This choice is preferable, as $\widetilde{Z}$ and $Z_{CG}$ determine two distinct types of cuts in the cell definition and their tesselation action is different: elements in $\widetilde{Z}$ translate the primitive CS-cell horizontally (plus some vertical shift along the pillar determined by $c_z$) while those in $Z_{CG}$ only vertically.

# C  Partial Bloch decomposition from representation theory

In this appendix we simplify the notation with $\mathcal{T}^L \to \mathcal{T}$.

## C.1  Induced representations

We remind the reader the definition of an induced representation. Let us assume the coset decomposition $g = t x$ with $g \in G$, $x \in X$ and $t$ in one of its transversal. Suppose we have a representation $\sigma$ of $X$ acting on a vector space $W$. Then, we may construct the induced representation of $G$ acting on the space $V$ made of functions

$$\mathbf{v} = \sum_t \mathbf{e}_t \otimes \mathbf{w}_t, \tag{C.1}$$

with $\mathbf{w}_t \in W$ for each $t$, from the action

$$(\mathrm{Ind}_X^G \sigma)_g (\mathbf{v}) = \sum_{t'} \mathbf{e}_{t_{gt'}} \, \sigma_{x_{gt'}} \cdot \mathbf{w}_{t'}, \tag{C.2}$$

where it holds $g\, t' = t_{gt'} x_{gt'}$.

It is known that the left regular representation of any subgroup $X$ of $G$, we denote it by $\mathcal{T}_X$, induces the left regular representation of the group [52]. Thus, if $\sigma = \mathcal{T}_X$, using Eq. (C.1) and Eq. (C.2), we have $\langle t\, x | \mathcal{T}_g \psi \rangle = \langle \bar{x}_{gt'} x | \varphi_{t'} \rangle$ where we used the braket notation $(\mathbf{e}_t, \mathbf{v}, \mathbf{w}) \to (|t\rangle, |\psi\rangle, |\varphi\rangle)$ and $t'$ is such that $t_{gt'} = t$. If, in addition, $X = C$ one has $x_{gt'} = l_{\mathbf{r} - \mathbf{r}_g, g}$, where $t = g_{\mathbf{r}}$ and $t' = g_{\mathbf{r} - \mathbf{r}_g}$.

An interesting property of induced representations, is that $\mathcal{T}$ can be written as the direct sum of the induced representations of the irreps $\{\sigma_\alpha\}_\alpha$ of $\mathcal{T}_X$:

$$\mathcal{T} = \bigoplus_\alpha \mathrm{Ind}_X^G (\sigma_\alpha^X)^{\oplus d_\alpha} \overset{def}{=} \bigoplus_\alpha \mathcal{T}_\alpha^{\oplus d_\alpha} \tag{C.3}$$

with $d_\alpha$ the dimension of the representation. We stress that the summands $\mathcal{T}_\alpha$ are not necessarily irreducible.

## C.2  Induction from two subgroups

Central subgroups, being abelian, have only 1D irreps $\chi_{\eta, y}^Y = e^{-i \boldsymbol{\eta} \cdot \mathbf{y}}$ with $\boldsymbol{\eta}$ parameterizing the dual group and $y \in Y$ having an associated vector $\mathbf{y}$ by abelianity. Considering only central subgroups that do not overlap with $C$, e.g. $Y = Q, \tilde{Z}$ we can make an additional step in the decomposition Eq. (C.3). In these cases, $\chi_{\eta, y}^Y \equiv e^{-i \boldsymbol{\eta} \cdot \mathbf{r}_y}$ where $\mathbf{r}_y$ is extracted from the decomposition Eq. (3) of $y$ and $\boldsymbol{\eta}$ lives in a Brillouin zone $BZ_Y$ defined by reciprocal vectors $\mathbf{G}_j$ satisfying $\exp(i \mathbf{G}_j \cdot \mathbf{r}_{y_k}) = \delta_{jk}$, with $\{y_k\}_k$ a generating set of $Y$.

We shall denote by $\chi_{\eta, g}^G$ the multiplicative representation $e^{-i \boldsymbol{\eta} \cdot \mathbf{r}_g}$ of $G$, trivial on $C$, which extends the domain of $\chi^Y$ from $Y$ to $G$. We observe that, since $\mathcal{T}_{\eta, y} \equiv \chi_{\eta, y}^Y$ then the *twisted* representation

$$\widetilde{\mathcal{T}}_{\eta, g} = \bar{\chi}_{\eta, g}^G \, \mathcal{T}_{\eta, g} \tag{C.4}$$

factors through the quotient by $Y$, i.e. it is the identity when evaluated on the subgroup $Y$. We show in Proposition 3.3 that the representation $\breve{\mathcal{T}}_\eta$ of $G/Y$ defined by $\breve{\mathcal{T}}_{\eta, gY} = \widetilde{\mathcal{T}}_{\eta, g}$ ($g \in G$) is (unitary-)equivalent to $\mathcal{T}_{G/Y}$. This fact allows to regard translations as a composition of an extra-cell cell shift, associated by $Y$, and an intra-cell one, associated by $G/Y$.

We stress that it is not possible to include $Z_{CG}$ in the groups $Y$. However, this fact is not relevant since we can use the entire group $C$ to refine the decomposition of $\mathcal{T}$. Consider Eq. (C.3) with $X = YC$ (a direct product of subgroups). We can write

$$\mathcal{T} = \mathrm{Ind}_{YC}^G (\mathcal{T}_Y \, \mathcal{T}_C) = \bigoplus_{\eta, \xi} \mathrm{Ind}_{YC}^G \left( \chi_\eta^Y \, \sigma_\xi^C \right)^{\oplus d_\xi}$$

$$\overset{def}{=} \bigoplus_{\eta, \xi} \chi_\eta^G \, \widetilde{\mathcal{T}}_{\eta \xi}^{\oplus d_\xi} \tag{C.5}$$

where $\mu = 1, \ldots, d_\xi$ labels the copies of the representation $\xi$ of $C$ and we defined the representation $\widetilde{\mathcal{T}}_{\eta\,\xi}$ that, in analogy of what was mentioned above about $\widetilde{\mathcal{T}}_\eta$, factors through the quotient by $Y$ and is associated with a representation $\widecheck{\mathcal{T}}_{\eta\,\xi}$ of $G/Y$ of dimension $d_\xi |G/(YC)|$.

Unfortunately, it is not possible to create a factored representation which would be equivalent to $\mathcal{T}_{G/(YC)}$, unless $\xi$ is the trivial representation. Indeed, with $\xi$ nontrivial it is not possible to extend $\sigma_\xi^C$ from $C$ to a representation of $G$. Instead, in the trivial case, immediately one defines $\sigma_{\xi=id,g}^G = \sigma_{\xi=id,c_g}^C = 1$ and we can state that $\widetilde{\mathcal{T}}_{\eta\xi=id,g} \sim \mathcal{T}_{G/(YC)}$.

Finally, we note that such decomposition of $\mathcal{T}$ in Eq. (C.5) is partial, that is, in principle the invariant subspaces appearing above could be further reduced. However, if the irreps of $G/Y$ are known, then the eigenspace decomposition of $\widetilde{\mathcal{T}}_{\eta,\xi,g}$, with $\xi$ generic, could be recovered as part of the full irreps decomposition of $\widetilde{\mathcal{T}}_\eta$ ($\sim \widetilde{\mathcal{T}}_{G/Y}$).

### C.3   Bloch eigenfunctions

The decomposition Eq. (C.5) can be associated to Bloch functions $\{|\mathcal{B}_{\eta\xi\mu}^{(n\nu)}\rangle\}_{n\nu}$ ($1 \leqslant n \leqslant |T_{YC}^{ab}|$, $1 \leqslant \nu \leqslant |d_\xi|$) that form the invariant dynamical subspace of $L^2(G)$ of Eq. (13). They can be defined as the product of functions

$$|\mathcal{B}_{\eta\xi\mu}^{(n\nu)}\rangle = |\varphi_n\rangle|\chi_\eta\rangle|\sigma_\xi^{\mu\nu}\rangle \tag{C.6}$$

To evaluate its components, we doubly coset-decompose $g = t_g y_g c_g$ where $t_g$ is an element of the transversal $T_{YC}^{ab} \subset T_n^{ab}$ comprising the abelian support within the cell associated with $G/(YC)$. $|\chi_\eta\rangle \in L^2(Y)$ evaluates as (we use the same notation of Eq. (1) here) $\langle z_g|\chi_\eta\rangle = \overline{\chi}_{\eta,z_g}$. $|\sigma_\xi^{\mu\nu}\rangle \in L^2(C)$ evaluates as $\langle c_g|\sigma_\xi^{\mu\nu}\rangle = \overline{\sigma}_{\xi,c_g}^{C,\mu\nu}$, where we use the basis $B_{\xi\mu}$ shown below Eq. (7); the set of functions $\{\varphi_n\}_{1 \leqslant n \leqslant |T_{YC}^{ab}|}$ form an orthonormal basis of $L^2(T_{YC}^{ab})$, they evaluate as $\langle t_g|\varphi_n\rangle = \varphi_n(t_g)$ and are analogous to the periodic part of the Bloch functions $u^{(n)}(\mathbf{r})$ commonly used in Euclidean crystals.

### C.4   Induction from $PC$

$P$ is the easiest subgroup to identify, therefore, we consider the decomposition along the group $PC$. Since in general there is no inclusion relation between $P$ and $\widetilde{Z}$, the invariant spaces identified in this decomposition can be different. From Eq. (C.3) we get

$$\mathcal{T} \;=\; \bigoplus_{\mathbf{k},\xi} \mathrm{Ind}_{PC}^G(\chi_\mathbf{k}^P \sigma_\xi) \overset{def}{=} \bigoplus_{\mathbf{k},\xi} \mathcal{T}_{\mathbf{k}\xi} \tag{C.7}$$

where $\mathbf{k}$ is a *standard* quasi-momentum in a toric $BZ$.

We might define the representation $\widetilde{T}_\mathbf{k}$ as above, however we can not state any equivalence with the representation of the quotient of $G$ over $P$ since $P$ is not necessarily normal. Therefore, such decompositions might not explicitly highlight all spectral degeneracies (cf. Appendix D.3).

# D   Spectra of the examples of Section 9

We shall assume $\kappa_\lambda = 1$ for simplicity and we lighten the notation with $\mathcal{T}^L \to \mathcal{T}$. In all examples the representation $\mathcal{T}_{\eta\omega}$ acts upon Bloch wavefunctions $|\mathcal{B}_{\eta\omega}^{(n)}q\rangle = |\varphi^{(n)}\rangle|\chi_\eta\rangle|\sigma_\omega\rangle$ (cf. Eq. (C.6)) and we choose a canonical basis for $L^2(T_{\widetilde{Z}C}^{ab})$, $\langle t|\varphi^{(n)}\rangle = \delta_{nt}$.

## D.1    $C = \mathbb{Z}_2$ and $n = 2$

## D.2    Hamiltonian analysis: Bloch decomposition along the primitive CS-supercell

We block-diagonalize the Hamiltonian using Eq. (13) with respect to the group $\widetilde{Z}C$ (see purple square in Fig. 3). The $pBZ$ vectors are $\boldsymbol{\eta} = \eta_1(1/2, 0) + \eta_2(0, 1/2)$   $(\eta_{1,2} \in [-\pi, \pi])$, where we set the lattice spacing to unity. Using the definition Eq. (C.2) we get

$$
\begin{aligned}
\mathcal{T}_{\boldsymbol{\eta}\omega,1}|\varphi\rangle &= \varphi_e|1\rangle + \chi_{\eta_1}\varphi_1|e\rangle + \varphi_2|12\rangle + \chi_{\eta_1}\varphi_{12}|2\rangle, \\
\mathcal{T}_{\boldsymbol{\eta}\omega,2}|\varphi\rangle &= \varphi_e|2\rangle + \chi_{\omega}\varphi_1|12\rangle + \chi_{\eta_2}\varphi_2|e\rangle + \chi_{\eta_2}\chi_{\omega}\varphi_{12}|1\rangle,
\end{aligned}
\tag{D.1}
$$

and

$$
\begin{aligned}
\widetilde{\mathcal{T}}_{\boldsymbol{\eta}\omega,1} = e^{i\eta_1/2}\mathcal{T}_{\boldsymbol{\eta}\omega,1} &= \begin{pmatrix} 0 & e^{-i\eta_1/2} & 0 & 0 \\ e^{i\eta_1/2} & 0 & 0 & 0 \\ 0 & 0 & 0 & e^{-i\eta_1/2} \\ 0 & 0 & e^{i\eta_1/2} & 0 \end{pmatrix}, \\
\widetilde{\mathcal{T}}_{\boldsymbol{\eta}\omega,2} = e^{i\eta_2/2}\mathcal{T}_{\boldsymbol{\eta}\omega,2} &= \begin{pmatrix} 0 & 0 & e^{-i\eta_1/2} & 0 \\ 0 & 0 & 0 & \omega e^{-i\eta_1/2} \\ e^{i\eta_1/2} & 0 & 0 & 0 \\ 0 & \omega e^{i\eta_1/2} & 0 & 0 \end{pmatrix},
\end{aligned}
\tag{D.2}
$$

where $|\varphi\rangle \in L^2(T_{\widetilde{Z}C}^{ab}))$, we neglected the diagonal action of $\mathcal{T}$ on $|\chi_{\boldsymbol{\eta}}\rangle|\sigma_{\omega}\rangle$ and the matrix degrees are organized, from innermost to outermost, as $(e, 1) \otimes (e, 2)$.

The Bloch Hamiltonian reads as

$$
\mathcal{H}_{\boldsymbol{\eta}\omega} = \begin{pmatrix} 0 & g_{\eta_1} & g_{\eta_2} & 0 \\ g_{\eta_1}^* & 0 & 0 & \omega g_{\eta_2} \\ g_{\eta_2}^* & 0 & 0 & g_{\eta_1} \\ 0 & \omega g_{\eta_2}^* & g_{\eta_1}^* & 0 \end{pmatrix}
\tag{D.3}
$$

with $g_\eta = 1 + e^{-i\eta}$.

The $\omega = +1$ sector controls the abelian states. Moreover, as commented below Eq. (C.5), on this eigenspace the associated representation $\widecheck{\mathcal{T}}_{\boldsymbol{\eta}+}$ is the regular representation of $G/(\widetilde{Z}C) = G/Z = \mathbb{Z}_2 \times \mathbb{Z}_2$. Thus, all translations can be diagonalized on the same basis to get the spectrum

$$
E_{\boldsymbol{\eta}+} = s_1|g_{\eta_1}| + s_2|g_{\eta_2}|,
\tag{D.4}
$$

where each choice of the pair of signs $s_j = \pm 1$ corresponds to a different band (see green bands in Fig. 6(row below)). The Bloch eigenvectors are associated to periodic plane waves on the group $\mathbb{Z}_2 \times \mathbb{Z}_2$ on top of an overall phase modulation $e^{i\boldsymbol{\eta}\cdot\mathbf{r}_z}$ on $\widetilde{Z}$.

The $\omega = -1$ sector retains the non-commutativity of the translations along the two directions, made evident by $\widetilde{\mathcal{T}}_{\boldsymbol{\eta}-,c} = -\mathbb{1} \neq \mathbb{1}$ (cf. Eq. (D.1)). According to Proposition 3.3, $\widecheck{\mathcal{T}}_{\boldsymbol{\eta}-}$ does not represent $G/Z$, instead its represents (unfaithfully) $G/\widetilde{Z} = D_8$. The structure of this group is well known: a regular representation has 5 distinct irreps, 4 of which are 1D (the abelian eigenspace), and one which is 2D and repeats twice. Thus, using the observation at the end of Appendix C.2 and reasoning by exclusion, we know that $\widecheck{\mathcal{T}}_{\boldsymbol{\eta}-}$ splits in 2 copies of a 2D irreps. Therefore, we expect $\mathcal{H}_{\boldsymbol{\eta}-}$ to have 2 distinct degenerate bands. Indeed, diagonalizing Eq. (D.3) one finds only one pair of energies,

$$
E_{\boldsymbol{\eta}-} = \pm\sqrt{|g_{\eta_2}|^2 + |g_{\eta_1}|^2},
\tag{D.5}
$$

depicted in pink in Fig. 6(row below).

### D.3  Hamiltonian analysis: Bloch decomposition along the standard unit CS-cell

This case is analyzed as before but, since the (standard) $BZ$ differs from the $pBZ$, we make the substitution $\boldsymbol{\eta} \to \boldsymbol{k}$ with $\boldsymbol{k} = k_1(1/2, 0) + k_2(0, 1)$  $(k_{1,2} \in [-\pi, \pi])$. Using again the definition Eq. (C.2), it holds

$$
\begin{aligned}
\mathcal{T}_{\boldsymbol{k}\omega,1}|\varphi\rangle &= \varphi_e|1\rangle + \chi_{k_1}\varphi_1|e\rangle, \\
\mathcal{T}_{\boldsymbol{k}\omega,2}|\varphi\rangle &= \chi_{k_2}\varphi_e|e\rangle + \chi_{k_2}\chi_\omega\varphi_1|1\rangle
\end{aligned}
\tag{D.6}
$$

and

$$
\widetilde{\mathcal{T}}_{\boldsymbol{k}\omega,1} = e^{ik_1/2}\mathcal{T}_{\boldsymbol{k}\omega,1} = \begin{pmatrix} 0 & e^{-ik_1/2} \\ e^{ik_1/2} & 0 \end{pmatrix},
$$

$$
\widetilde{\mathcal{T}}_{\boldsymbol{k}\omega,2} = e^{ik_2}\mathcal{T}_{\boldsymbol{k}\omega,2} = \begin{pmatrix} 1 & 0 \\ 0 & \omega \end{pmatrix},
\tag{D.7}
$$

where the matrix degrees are organized as $(e, 1)$.

The Bloch Hamiltonian reads as

$$
\mathcal{H}_{\boldsymbol{k}\omega} = \begin{pmatrix} f_{k_2} & g_{k_1} \\ g_{k_1}^* & \omega f_{k_2} \end{pmatrix}
\tag{D.8}
$$

with $f_k = 2\cos(k)$ and $g_k = 1 + e^{-ik}$.

The bands spectrum is easily computed as

$$
E_{\mathbf{k}+} = \pm|g_{k_1}| + f_{k_2}
\tag{D.9}
$$

$$
E_{\mathbf{k}-} = \pm\sqrt{|g_{k_1}|^2 + f_{k_2}^2}
\tag{D.10}
$$

Notice that this spectrum is obtained upon unfolding the $pBZ$ along the direction $\lambda = 2$. The band, whose eigenvectors have the modulation $\pm 1$ within the unit cell along such direction, unfolds in the $BZ$ to momenta $\mathbf{k} = (\eta_1, \eta_2 + \pi)$ (the red arrows in Fig. 6 show the reversed folding procedure). Hence, we recover Eq. (D.9) from Eqs. (D.4) and (D.5) upon the substitution $\pm|g_{\eta_2}| \to f_{k_2}$.

It is interesting to notice that the band degeneracy of $E_{\eta-}$ is lost in passing to $E_{\mathbf{k}-}$. Analyzing the Hamiltonian with the standard unit CS-cell may hide structural degeneracies that can be confused with accidental ones. The analysis through the primitive CS-cells highlights better such degeneracies.

Finally, concerning the $\omega = +1$ sector, we may operate an even further unfolding to a cell comprising only one pillar. We would get $E_{\mathbf{k}+}^{square} = f_{k_2} + f_{k_1}$ as expected from an isotropic nearest-neighbor tight binding model on a square lattice. For the abelian states the standard unit CS-cell is a redundant structure.

### D.3.1  PBC using the group $P$

As mentioned in Section 7, choosing PBC groups that are not in any $\widetilde{Z}_i$ leads to non-Cayley-crystals. Suppose we choose such group to be $N = \langle 1^2, 2^3 \rangle_{ab}$. It holds $P > N \nsubseteq \widetilde{Z}$. If we add the relations of $N$ into the presentation of $G$, in Eq. (14), trying to construct $G/P$ we would soon realize some contradiction. For instance, by the relations of $G$ it holds $\bar{1}\bar{2}1 = \bar{2}c$ and it follows $\bar{1}\bar{2}^31 = (\bar{2}c)^3 = \bar{2}^3 c$. However, since $2^3 = \bar{2}^3 = e$ by the relations of $N$, that equation leads to the unacceptable identification $e = c$, which would reduce the number of points of the tentative group to 6, instead of 12 as suggested by the transversal of $N$ in $G$. However, we

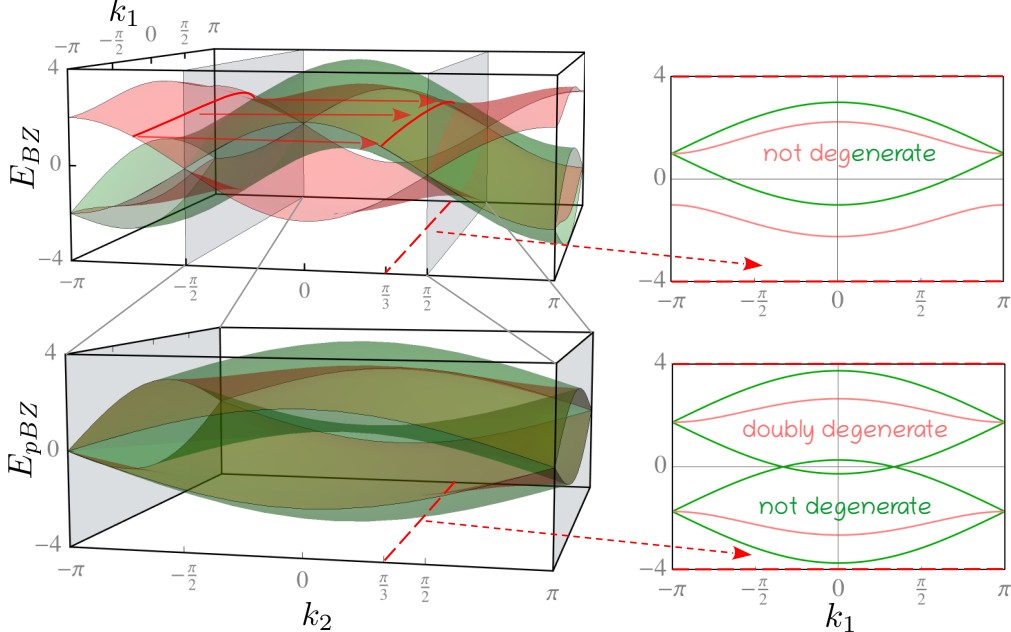

Figure 6: (left) Spectrum $E_{\mathbf{k}}$ of $\mathcal{H}$ in the standard $BZ$ related to the group $P$ (row above) and the primitive folded one (row below). The abelian bands are shown in green while the non-abelian ones in pink. The folding planes along $k_2$ in going from one $BZ$ to the other are shown in gray; the red arrows show how the red band gets folded in going to the $pBZ$. (right) cut section of the spectrum along the dashed red line (along $k_1$) in the left plots. The non-abelian bands get doubly degenerate in the $pBZ$.

cannot ignore that the transversal of $N$ itself is a periodic lattice and the Hamiltonian could be diagonalized using the standard unit CS-cell. The abelian sector does not present any novelty with respect to the valid PBC case. The non-abelian sector instead describes a particle in a magnetic flux on a $2 \times 3$ toric lattice. Importantly the flux threading the whole lattice along the vertical direction equals $3\pi(\Phi_0/2\pi)$, not a multiple of the flux quanta. As a result the eigenstates, which are plane waves in each direction, present a $\pi$ shift in momentum along the direction 2 to adjust to the Born-von Karman boundary condition [6,77]. Notice that the set of momenta $k_2$ does not include the point $k_2 = 0$.

## D.4    $C = \mathbb{Z}_3$ **and** $n = 2$

We shall analyze only the two opposite-extreme cases in terms of the non-triviality of the powers $a_i$, that is, case 'a' and 'd'.

### D.4.1    Case 'a'. Brief Hamiltonian analysis

This case features a central embedding of $C$ and its analysis is analogous to that of the single CS-crystal analyzed in the previous example. The bands for the non-trivial $C$-irreps, produced using the standard CS-cell, are those of the Hofstädter butterfly at 1/3 of the flux quantum [1]. They are gapped and present topological properties.

### D.4.2 Case 'd'. Hamiltonian analysis: primitive unit CS-cell

One motivation for a closer look at this case comes from its tilted primitive unit CS-cell (see the purple area in Fig. 4). The *pBZ* vectors have the form $\boldsymbol{\eta} = \eta_1(1,1)/2 + \eta_2(-1,1)/2$ ($\eta_{1,2} \in [-\pi, \pi]$), where we set the lattice spacing to unity. We get

$$
\begin{aligned}
\mathcal{T}_{\boldsymbol{\eta}\omega,1}|\varphi\rangle &= \omega\bar{\chi}_{\eta_2}\varphi_e|2\rangle + \omega^2\chi_{\eta_1}\varphi_2|e\rangle, \\
\mathcal{T}_{\boldsymbol{\eta}\omega,2}|\varphi\rangle &= \varphi_e|2\rangle + \chi_{\eta_1}\chi_{\eta_2}\varphi_2|e\rangle
\end{aligned}
\tag{D.11}
$$

and

$$
\begin{aligned}
\widetilde{\mathcal{T}}_{\boldsymbol{\eta}\omega,1} &= e^{i(\eta_1-\eta_2)/2}\mathcal{T}_{\boldsymbol{\eta}\omega,1} = \begin{pmatrix} 0 & \omega^2 e^{-i(\eta_1+\eta_2)/2} \\ \omega e^{i(\eta_1+\eta_2)/2} & 0 \end{pmatrix}, \\
\widetilde{\mathcal{T}}_{\boldsymbol{\eta}\omega,2} &= e^{i(\eta_1+\eta_2)/2}\mathcal{T}_{\boldsymbol{\eta}\omega,2} = \begin{pmatrix} 0 & e^{-i(\eta_1+\eta_2)/2} \\ e^{i(\eta_1+\eta_2)/2} & 0 \end{pmatrix},
\end{aligned}
\tag{D.12}
$$

where the matrix degrees are organized according to the vector $(e, 2)$. One can check that the associated representation $\widecheck{\mathcal{T}}_{\boldsymbol{\eta},\omega}$ represents (unfaithfully) the group $G/\widetilde{Z} = S_3$.

The Bloch Hamiltonian reads as

$$
\mathcal{H}_{\boldsymbol{\eta}\omega} = \begin{pmatrix} 0 & f_{\boldsymbol{\eta},\omega} \\ f^*_{\boldsymbol{\eta},\omega} & 0 \end{pmatrix}
\tag{D.13}
$$

with $f_{\boldsymbol{\eta},\omega} = \omega^2(e^{-i\eta_1} + e^{-i\eta_2}) + 1 + e^{-i(\eta_1+\eta_2)}$.

The chiral symmetric spectrum is easily computed as $E_{\boldsymbol{\eta},\omega} = \pm|f_{\boldsymbol{\eta},\omega}|$ and is found to be gapless in all three $\omega$-sectors. A part from the property $f^*_{\boldsymbol{\eta},\omega} = f_{-\boldsymbol{\eta},\omega*}$, implied by the time reversal symmetry of the whole lattice, the spectrum obeys $|E^i_{\boldsymbol{\eta},\omega}| = |E^j_{\boldsymbol{\eta},\omega*}|$ $(i, j = 1, 2)$, which implies band degeneracy if we had put the two non-trivial $C$-irreps in direct sum. Here, we remark a difference with respect to the case in Section 9.1 where the degeneracy appeared *within* the non-trivial irrep. This difference is of no concern, since the theory requires only the representation $\widecheck{\mathcal{T}}_{\boldsymbol{\eta}}$ to be *regular* while $\widecheck{\mathcal{T}}_{\boldsymbol{\eta},\omega}$ does not have to, see Proposition 3.3. As a consequence, since the represented group $S_3$ has one irrep of dimension two, we get the band degeneracy only if we put in direct sum the irreps of $C$ shown in Eq. (D.12).

## E Theorems

In the following theorems we denote by $\prod_{a=1}^n c_a$ the string $c_1 c_2 \cdots$ and by $\prod'^n_{a=1} c_a$ the one with reversed order $c_n c_{n-1}\cdots$. We use the notation $c^{(g)} = \bar{g}\, c\, g$ for conjugations where the overbar denotes inversion.

We shall consider the basis for the commutator group $F'_n$ of the free group $F_n$ given by [78]

$$
\mathcal{F}'_n = \{c_{ij}^{(g_\mathbf{r})} : 1 \leqslant i < j \leqslant n, g_\mathbf{r} \in T^{ab}_n\}
\tag{E.1}
$$

It is a consequence of the universality of free groups that $C$ is a quotient group of $F'$. Therefore, $\mathcal{F}'_n$ can be used as a (redundant) basis for $C$ in a CS-lattice.

### E.1 Group $P$ and standard CS-lattice periodicity

**Proposition 1.1.** *The links defined in Eq. (4) can be expressed explicitly in the basis $\mathcal{F}'_n$ (see Eq. (E.1)):*

- *Assuming $\lambda = i \in \{1, 2, \ldots, n\}$, we have*

$$l_{\mathbf{r},i} = \prod_{j=1}^{i-1}{}' \prod_{p=0}^{|r_j|-1} \theta(r_j) \bar{c}_{ji}^{\left(j^p g_{\mathbf{r}_{\downarrow j+1}}\right)} + \theta(-r_j) c_{ji}^{\left(\bar{j}^{p+1} g_{\mathbf{r}_{\downarrow j+1}}\right)} \tag{E.2}$$

  *where we make use of the Heaviside $\theta$ function and we denote by $\mathbf{r}_{\downarrow m}$ the vector that retains all values of the components of $\mathbf{r}$ down to the $m$-th one while the other ones are set to zero.*

- *For a generic $\lambda$, backward links are computed from the forward ones as*

$$l_{\mathbf{r},\bar{\lambda}} = \overline{l_{\mathbf{r}-\mathbf{r}_\lambda, \lambda}} \tag{E.3}$$

- *For a generic $\lambda$, an expression in terms of the generators $\{1, 2, \ldots, n\}$ is obtained by combining Eq. (E.2) and the recursive formula*

$$l_{\mathbf{r},\lambda} = l_{\mathbf{r}+\lambda^{(RM)}, \lambda^{(L)}} l_{\mathbf{r}, \lambda^{(RM)}} \tag{E.4}$$

  *where we split $\lambda = \lambda^{(L)} \lambda^{(RM)}$ to highlight its rightmost (RM) element (taken with power one) and the leftover (L) part.*

*Proof.* We recall the definition of the links from Eq. (4): $l_{\mathbf{r},i} = \overline{\prod_{j=1}^{n} j^{r_j + \delta_{ij}}} \, i \left( \prod_{j=1}^{n} j^{r_j} \right)$. Let us focus on the first terms $x = i \left( \prod_{j=1}^{n} j^{r_j} \right)$. Moving $i$ on the right of its adjacent $j$-term one at a time until it meets the term $j \equiv i$ we get $x = 1^{r_1} i \, \bar{c}_{1^{r_1} i} \left( \prod_{j=2}^{n} j^{r_j} \right) = 1^{r_1} i \left( \prod_{j=2}^{n} j^{r_j} \right) \bar{c}_{1^{r_1} i}^{\left( \prod_{j=2}^{n} j^{r_j} \right)}$ $\rightarrow \left( \prod_{j=1}^{n} j^{r_j + \delta_{ji}} \right) \prod_{j=1}^{i-1}{}' \, \bar{c}_{j^{r_j} i}^{\left( \prod_{k=j+1}^{n} k^{r_k} \right)}$. Therefore,

$$l_{\mathbf{r},i} = \prod_{j=1}^{i-1}{}' \, \bar{c}_{j^{r_j} i}^{\left( \prod_{k=j+1}^{n} k^{r_k} \right)} \tag{}$$

.

If $r_j > 0$ We can decompose the commutators in the last product along the generator $j$ using the identity

$$c_{j^r i} = c_{ji}^{(j^{r-1})} c_{j^{r-1} i} \tag{E.5}$$

multiple times till exhaustion to get $c_{j^r i} = \prod_{p=0}^{r_j-1}{}' c_{ji}^{(j^p)}$. After substitution of this expression into the one for $l$, and using the definition of $\mathbf{r}_{\downarrow m}$ we get Eq. (E.2) (assuming $r_j > 0$). The case with $r_j < 0$ is very similar. The decomposition above still holds tih but with $j$ replaces with $\bar{j}$. We get the identity $c_{\bar{j}^r i} = \prod_{p=0}^{|r_j|-1}{}' c_{\bar{j} i}^{(\bar{j}^p)}$. Since $c_{\bar{j} i} = \bar{c}_{ji}^{(\bar{j})}$ we can rewrite the previous identity as $c_{\bar{j}^r i} = \prod_{p=0}^{|r_j|-1}{}' \bar{c}_{ji}^{(\bar{j}^{p+1})}$. Thus we lend to the full result of Eq. (E.2).

If the shift is backward, $\bar{\lambda}$, the formula Eq. (E.3) is easily retrieved by noticing, from *Eq.* (4), that $\overline{l_{\mathbf{r},\lambda}} = \overline{g_{\mathbf{r}}} \, \bar{\lambda} \, g_{\mathbf{r}+\mathbf{r}_\lambda} = l_{\mathbf{r}+\mathbf{r}_\lambda, \bar{\lambda}}$.

The recursive formula is easily obtained using the definition of links in Eq. (4) and splitting the central $\lambda$ element in two parts. $\square$

**Corollary 1.2.** *At fixed direction $i$ the related links can be constructed recursively dimension by dimension on the abelian support, speeding up the computation, by means of*

$$l_{\mathbf{r}_{\downarrow m}, i} = l_{\mathbf{r}_{\downarrow m+1}, i} \prod_{p=0}^{r_m-1} c_{mi}^{\left( m^p g_{\mathbf{r}_{\downarrow m+1}} \right)} \quad (i > m \geqslant 1) \tag{E.6}$$

*where it holds*

$$l_{\mathbf{r}_{\downarrow m}, i} = e \quad (i \leqslant m). \tag{E.7}$$

**Lemma 1.3.** *Every element of $C$ can be written as a product of links.*

*Proof.* Consider Eq. (4) substituting $\lambda$ with a generator $c$ of $C$. We get $l_{\mathbf{r},c} = \overline{g_{\mathbf{r}}} \, c \, g_{\mathbf{r}}$ and, in particular,

$$c = l_{\mathbf{0},c}.$$

We can now express $c$ explicitly in terms of the CS-lattice generators $\Lambda$ (which generate $G$ as well) as $c = \prod_k \lambda_k$. Thus, in analogy with the recursive formula Eq. (E.4), we can write $c = l_{\mathbf{0}, \prod_k \lambda_k} = \prod_k l_{\mathbf{r}_k, \lambda_k}$ for a certain set of positions $\mathbf{r}_k$ which we do not need to specify.  □

**Proposition 1.4.** *For each group $G$ with finite commutator group $C$ there exist a vector $v$ and a matrix $V$ with positive integer elements with $1 \leqslant v_i, V_{ij} \leqslant |C|$, $(1 \leqslant i, j \leqslant n)$ such that*

a) $c_i{}^{V_{ij}}{}_j = e \qquad i, j = 1, \ldots n$

b) $i^{v_i} \in \mathcal{C}_G(C) \qquad i = 1, \ldots n$

*where $\mathcal{C}_G(C)$ is the centralizer of $G$ in $G$.*

  *The minimal possible values of $v_i, V_{ij}$ can be found with an algorithm that does not require* a priori *knowledge of the center group and $\mathcal{C}_G(C)$.*

*Proof.* These properties are a consequence of the finiteness of the group $C$. Indeed, by finiteness and the pigeonhole principle for each $i$ and $j$ there exist positive integers $1 \leqslant x^*, y^* \leqslant |C|$ s.t.

$$c_i{}^{x^* + y^*}{}_j = c_i{}^{y^*}{}_j$$

i.e. one element of the group $|C|$ must repeat when scrolling through the set $\{c_i{}^x{}_j\}_x$ from $x = 0$ to increasing values. By writing the commutator explicitly in the equation above, simple algebraic simplifications lead to $c_i{}^{x^*}{}_j = e$. Then one sets $V_{ij} = x^*$. The scrolling procedure is important to guarantee that $x^*$ is the minimum periodicity one can get. This proves point a).

  Consider now a generating set of $C$. It embeds into $G$ as a subset $S$ whose elements are products of commutators. We use the same procedure as above to find, for each $s \in S$, the element in the set $M_s = \{s^{(i^{x_s})}\}_{x_s}$ for which $s^{(i^{x_s})} = s$. Finally, one defines the quantity $v_i = \text{m.c.m.} \{x_s^*\}_{s \in S}$ which happens to satisfy point b). Clearly, since $i^{v_i}$ commutes with all generators of $C \leqslant G$ it is in $\mathcal{C}_G(C)$. As with the definition of $V_{ij}$, $v_i$ is minimal; moreover, it is independent on the specific chosen set of generators of $C$ and its embedding. Indeed, if we had found another $v' \geqslant v$ (we omit the labels here) with a set $T \neq S$ then we would have $t^{(i^{v'})} = t$. However, since $T$ is generated by $S$, we have also $t^{(i^v)} = t$. Thus, since $v'$ was supposed to be minimal, it must hold $v' = v$.  □

**Theorem 1.5.** *(Periodicity of the infinite CS-crystal)*

1. *The periodicity of the infinite CS-crystal, moving along the abelian support, is determined by the group*

$$P = X \cap \mathcal{C}_G(C),$$

$$X = \{ g_{\mathbf{m}} \in T_n^{ab} : [\prod_{j=1}^{k-1} j^{m_j}, k] = e,$$

$$m_j \in \mathbb{Z}, \ k \in \{1, 2, \ldots, n\}\} \tag{E.8}$$

*where $\mathcal{C}_G(C)$ is the centralizer of $C$ in $G$ and is independent from the set of the crystal generators $\Lambda$.*

2. *$P$ is isomorphic to $\mathbb{Z}^n$.*

3. *P always contains the non-trivial subgroup:*

$$P_{\text{straight}} = \langle \{ i^{m_i} : m_i = \text{m.c.m.} \left( \{ V_{ij} | n \geqslant j > i \} \cup \{ v_i \} \right),$$
$$i \in \{1, \ldots, n\} \} \rangle. \tag{E.9}$$

*Proof.* Firstly we will prove point 1 supposing $\Lambda = \mathcal{F}_n = \{1, 2, \ldots, n\}$. Let us define $\mathbf{r}' = \mathbf{r} + \mathbf{m}$, with $\mathbf{m}$ a vector upon which

$$l_{\mathbf{r},i} = l_{\mathbf{r}',i}.$$

Using the definition of links in Eq. (4) and evaluating the formula above at $\mathbf{r} = \mathbf{0}$ one gets easily

$$[\prod_{j=1}^{i-1} j^m, i] = e. \tag{E.10}$$

Consider now $g_{\mathbf{r}'} = \prod_{j=1}^{n} j^{r+m}$ appearing in the definition of $l_{\mathbf{r}',i}$. By Eq. (E.10) we have $[1^m, 2] = e$, thus $g_{\mathbf{r}'} = 1^r 2^r 1^m 2^m \prod_{j=3}^{n} j^{r+m}$. As a second step, setting $i = 3$ in Eq. (E.10), we get $g_{\mathbf{r}'} = 1^r 2^r 3^r 1^m 2^m 3^m \prod_{j=4}^{n} j^{r+m}$. Repeating the steps up to $i = n - 1$ we land to the equation

$$g_{\mathbf{r}'} = \left( \prod_{j=1}^{n} j^r \right) \left( \prod_{j=1}^{n} j^m \right). \tag{E.11}$$

Since we can factorize analogously also the other product in $l_{\mathbf{r}',i}$, we get

$$l_{\mathbf{r}',i} = \overline{\left( \prod_{j=1}^{n} j^m \right)} \; \overline{\left( \prod_{j=1}^{n} j^{r+\delta_{ij}} \right)} i \left( \prod_{j=1}^{n} j^r \right) \left( \prod_{j=1}^{n} j^m \right)$$
$$= l_{\mathbf{r},i}^{(g_{\mathbf{m}})}, \tag{E.12}$$

hence, by $l_{\mathbf{r}',i} = l_{\mathbf{r},i}$ and Lemma 1.3 we have the additional constraint $g_{\mathbf{m}} \in \mathcal{C}_G(C)$. Thus we conclude that $g_{\mathbf{m}} \in P$.

To prove any element in $P$ produce periodicity, we can use again Eq. (E.12) that, together with $g_{\mathbf{m}} \in \mathcal{C}_G(C)$, implies $l_{\mathbf{r}',i} = l_{\mathbf{r},i}$.

Now, we have to show that the set $P$ is a group. First of all, it is closed under multiplication. Indeed, consider two elements of $P$, $a = g_{\mathbf{a}}$ and $b = g_{\mathbf{b}}$. Define $m_{\uparrow k} = \prod_{j=1}^{k} j^m$. Then we have

$$ab = a_{\uparrow n} b_{\uparrow n} = \left( \prod_{j=1}^{n} j^a \right) \prod_{j=1}^{n} j^b$$
$$= \left( \prod_{j=1}^{n-1} j^a \right) n^a \left( \prod_{j=1}^{n-1} j^b \right) n^b$$
$$= \left( \prod_{j=1}^{n-1} j^a \right) \left( \prod_{j=1}^{n-1} j^b \right) n^{a+b} = a_{\uparrow n-1} b_{\uparrow n-1} n^{a+b} \tag{E.13}$$

where we used Eq. (E.10) with $k = n$. The procedure can be continued till exhaustion in the subscript of $a_{\uparrow k} b_{\uparrow k}$ producing finally $ab = \prod_{j=1}^{n} j^{a+b}$. That every element has an inverse can

be seen similarly:

$$
\begin{aligned}
\bar{a} = \bar{a}_{\uparrow n} &= \left( \prod_{j=1}^{n}{}' \, \bar{j}^a \right) = n^{-m} \left( \prod_{j=1}^{n-1}{}' \, \bar{j}^a \right) \\
&= n^{-m} \overline{\prod_{j=1}^{n-1} j^a} = \left( \overline{\prod_{j=1}^{n-1} j^a} \right) n^{-m} = \bar{a}_{\uparrow n-1} n^{-m} \\
&= \prod_{j=1}^{n} j^{-a}
\end{aligned}
\tag{E.14}
$$

where again we used Eq. (E.10) in the third-to-last step, and the recursion in the last one. Obviously $P$ contain the trivial element, $e$, and the point 1 is proved with a restriction on the set $\Lambda$. To release this constraint, one can use recursively Eq. (E.4) to decompose $\lambda$ along the set $\mathcal{F}_n$ and see that the periodicity with the latter set implies periodicity on the new set $\Lambda$. The opposite implication can be seen in the same way since $\Lambda$ is a generating set and we can express $\mathcal{F}_n$ using that set.

To prove point 2 we observe that the set of vectors $\mathbf{m}$ such that $[\prod_{j=1}^{k-1} j^{m_j}, i] = e$ for all generators $x^k$ forms a subgroup $M$ of $\mathbb{Z}^n$. It follows by Nielsen–Schreier theorem that $M \cong \mathbb{Z}^n$. Hence, since every vector is in one-to-one correspondence with the elements of $P$ and we have the homomorphism $\mathbf{m}_{ab} = \mathbf{m}_a + \mathbf{m}_b$, then the groups are isomorphic. Thus, point 2 is proved.

It is straightforward to verify that $P_{\text{straight}} \leqslant P$ and is non-trivial. If a power $i^m$ is in the group $X$ (see Eq. (E.8)) then its minimal exponent is $m = \text{m.c.m.} \{V_{ij} | 1 \leqslant j < i\}$ ($V_{ij}$ is defined in Proposition 1.4). The additional requirement to be in the centralizer implies that $m \propto v_i$, defined in Proposition 1.4, which is again a minimal requirement. The non-triviality comes from the fact that the numbers $V_{ij}$ and $v_i$ are finite as shown in Proposition 1.4. $\qquad\square$

**Lemma 1.6.** *If $p \in P$ then $t\,p = (t\,p)^{ab}$ for any $t \in T_n^{ab}$.*

*Proof.* We first prove the right implication. We have the trivial equalities $t\,p = \left( \prod_{i=1}^{n} i^{t_i} \right) \prod_{i=1}^{n} i^{p_i} = (\prod i^t) \overline{\prod}' i^{-p}$, where in the last step we removed the obvious subscripts for better readability. By Eq. (E.14) we may rework the last expression to get $t\,p = (\prod i^t) \overline{\prod i^{-p}} = \overline{\prod i^{-p} \left( \prod' i^{-t} \right)}$. Finally, by applying repeatedly the commuting properties of the elements of $P$ at each $k = 1, \dots, n$, see its definition in Eq. (E.8) we get

$$
\begin{aligned}
t\,p &= \overline{\left( \prod_{i=1}^{n-1} i^{-p} \right) \bar{n}^{p+t} \left( \prod_{i=1}^{n-1}{}' \, i^{-t} \right)} \\
&= \bar{n}^{p+t} \overline{\left( \prod_{i=1}^{n-1} i^{-p} \right)} \, \prod_{i=1}^{n-1}{}' \, i^{-t} = \overline{\prod_{i=1}^{n}{}' \, \bar{i}^{p+t}} \\
&= \prod_{i=1}^{n} i^{p+t} = (t\,p)^{ab}.
\end{aligned}
\tag{E.15}
$$

To prove the converse we can take $t = i$. Upon explicit substitution of $p$ into $i\,p = (i\,p)^{ab}$, this condition becomes equivalent to $[\prod_{j=1}^{k-1} j^{m_j}, i] = e$. $\qquad\square$

**Lemma 1.7.** *(Cable formula for PBC clusters)*
*Let $I = G/N$ be an integral of $C$ where $G$ is a maximal integral and $N$ a valid PBC. Consider the CS-lattice of $I$ embedded in $G$, that is, $I$ identified with a transversal of $N$ in $G$ whose abelian*

*support is made of adjacent points and convex. The cable formula at the boundary of the CS-lattice of I reads as*

$$c_{\mathbf{r},\lambda}^{I}(c^{I}) = l_{\mathbf{r},\bar{n}_{\mathbf{r},\lambda}\lambda} \, c^{I} \tag{E.16}$$

*with $\lambda$ is the shift, $\mathbf{r}$ lies inside the abelian support of I, $\mathbf{r}+\mathbf{r}_\lambda$ lies outside (inside that of G) and $n_{\mathbf{r},\lambda}$, a generator of N, is such that $\mathbf{r}+\mathbf{r}_\lambda - \mathbf{r}_{n_{\mathbf{r},\lambda}}$ lies inside.*

*Moreover, the twist factor defined as $c_{\mathbf{r},\lambda}^{I} = c_{\mathbf{r},\lambda}^{twist} \, c_{\mathbf{r},\lambda}$, with $c_{\mathbf{r},\lambda}$ satisfying the cable formula Eq. (5) of G, amounts to*

$$c_{\mathbf{r},\lambda}^{twist} = l_{\mathbf{r}+\mathbf{r}_\lambda, \bar{n}_{\mathbf{r},\lambda}}. \tag{E.17}$$

*Proof.* The quotient in $I = G/N$ implies the equivalence $ng^{ab}c \sim g^{ab}c$ in $G$ for any $n \in N$. The cable along a translation by $\lambda$ in $G$ is obtained from $g_{new} = \lambda \, g_{old}$ that implies formula Eq. (5) (in the following "old" refers to object on a certain pillar and "new" to those on the pillar advanced by $\mathbf{r}_\lambda$). Within $I$, if $g_{\mathbf{r}+\mathbf{r}_\lambda}^{ab}$ happens to lie outside the abelian support we can write $g_{new} = n_{\mathbf{r},\lambda} g_{new}^{I}$ with $n_{\mathbf{r},\lambda} \in N$ (we suppress this subscript in the following) and $g_{new}^{I}$ inside the support. Thus, we get $ng_{new}^{I} = \lambda \, g_{old}$. Using the decomposition Eq. (3) we get $g_{\mathbf{r}+\mathbf{r}_g-\mathbf{r}_n}^{ab} c_{new}^{I} = \bar{n}\lambda g_{\mathbf{r}}^{ab} c_{old}$ where $\mathbf{r}_n = n^{ab}$, whence, by using the definition of links in Eq. (4) and changing the notation for $c_{new/old}$, we obtain Eq. (E.16). The twist factor is recovered, by observing that $c_{old}^{I} = c_{old}$ and applying the splitting formula Eq. (E.4) in Eq. (E.16). $\qquad\square$

## E.2   Central groups $Z$ and $Q$, centralizer of $C$ and their properties

**Corollary 2.1 of Proposition 1.4**

*The center group of G is non-trivial and contains $\mathbb{Z}^n$.*

*Proof.* The elements of $Z_{\text{straight}} = \{i^{V_i} : V_i = \text{m.c.m.} \{V_{ij}\}_{j=1,2,\dots,n}, i = 1,2,\dots,n\}$ are central, by the definition of the $V_{ij}$, and generate a free abelian group. $\qquad\square$

**Proposition 2.2.** *(Center factorization and rank)*

*The center of G can be written as the direct product of subgroups $Z = \widetilde{Z}Z_{CG}$ where $\widetilde{Z} \cong \mathbb{Z}^n$, $Z_{CG} = Z \cap C$ and $\widetilde{Z} \cap Z_{CG}$. It holds $\text{rank}(Z) = n + \text{rank}(Z_{CG})$.*

*Proof.* We may embed $Z$ in $G^{ab}$ with the canonical projection

$$\phi : G \ni g \mapsto g\, C \in G^{ab}. \tag{E.18}$$

Using such homomorphism and the first isomorphism theorem we have $\phi(Z) \cong Z/(\text{Ker}(\phi)\cap Z) = Z/Z_{CG}$. It is a known fact that for free abelian groups $\text{rank}(X) \geqslant \text{rank}(Y)$ if $X \geqslant Y$. In our case we have $n = \text{rank}(G^{ab}) \geqslant \text{rank}(\phi(Z)) \geqslant \text{rank}(\phi(Z_{\text{straight}})) = \text{rank}(Z_{\text{straight}}) = n$, where $Z_{\text{straight}}$ is the group defined in Appendix E.2. This implies $\text{rank}(\phi(Z)) = n$ and, since $\phi(Z)$ is abelian, $\phi(Z) \cong \mathbb{Z}^n$. We make the observation that $Z_{CG}$ is the torsion group of $Z$. Indeed, $C$ is finite by assumption and all elements of $Z\backslash Z_{CG}$ have a finite abelianized component, thus having infinite order. Therefore, by the classification of finitely presented abelian groups [39] we have the decomposition $Z = \widetilde{Z}Z_{CG}$, where $\widetilde{Z} \cong Z/Z_{CG} \cong \phi(Z) \cong \mathbb{Z}^n$. The second claim is a simple corollary of the first claim.

$\qquad\square$

**Proposition 2.3.** $q \in Q = P \cap Z$ *if and only if* $q\,t = t\,q = (q\,t)^{ab}$ *for any* $t \in T_n^{ab}$.

*Proof.* The right implication follows straightforwardly from Lemma 1.6 and the additional requirement that $q \in \widetilde{Z}$. The left implication follows since the first equality in the assumption implies $q \in Z \leqslant \mathcal{C}_G(C)$ whereas the second one that implies $q \in X$, with the group $X$ as in Eq. (E.8). $\qquad\square$

**Corollary 2.4.** *(Relation between Q and Z)*
  *The only non-trivial maximal group contained in the set $Z \cap T_n^{ab}$ is Q.*

**Corollary 2.5.** *(Group Q and Wilson-loops)*
  *It holds $l_{\mathbf{r},q} = e$ for any $\mathbf{r} \in \mathbb{Z}^n$ and $q \in Q$. In particular, the C-valued Wilson-loop around a standard CS-supercell vanishes.*

*Proof.* The latter statement can be proved using the recursive formula in Eq. (E.4) with $\lambda$ equal to the $C$-valued Wilson-loop string. $\qquad\square$

**Proposition 2.6.** *(Group Q and valid PBCs)*

  (a)  *The subgroups of $Q = P \cap Z$ are all and only the standard PBCs.*

  (b)  *Q is isomorphic to $\mathbb{Z}^n$.*

*Proof.* Consider Lemma 1.7 and, in particular, the twisting factor at the boundary in Eq. (E.17), $c_{\mathbf{r},\lambda}^{twist}$. If $n_{\mathbf{r},\lambda} \in Q$, the factor is trivial by Corollary 2.5. Moreover, since $Q \leqslant P$, the lattice cut produced by $n_{\mathbf{r},\lambda}$ does not go through a standard CS-cell. Thus elements of $Q$ generate valid PBCs. The other implication stems directly from the definition of $Q$. Consider a valid PBC that does not cut through a periodic CS-cell, then the the PBC element associated to it must be in $P$ by Theorem 1.5. Moreover, since the PBC is a valid one, the element must also be in $\widetilde{Z}$, whence it is in $Q$.

  $Q$ is a subgroup of $P$ therefore, by the Nielsen-Schreier theorem, is also free abelian. At the same time $Q \geqslant P_{\text{straight}} \cap Z_{\text{straight}}$ where the first group is defined in Theorem 1.5 and the second in Appendix E.2. Such intersection is nontrivial since for each generator $i = 1, 2, \ldots, n$ we can find a finite m.c.m. of the respective powers in the first and second group generators and obtain and element in $Q$. This intersection is clearly isomorphic to $\mathbb{Z}^n$. Using the same argument used in Proposition 2.2, with the chain of inclusion $P \geqslant Q \geqslant P_{\text{straight}} \cap Z_{\text{straight}}$, one proves that $Q$ is isomorphic to $\mathbb{Z}^n$.

  $\qquad\square$

**Proposition 2.7.** *(Presentation of $\mathcal{C}_G(C)$)*
  *Given a maximal integral G of a cyclic group C, then $\mathcal{C}_G(C) = C\,T$ is abelian, with $T = \{\prod_{i=1}^n i^{s_i} : \prod_{i=1}^n a_i^{s_i} = 1\,(\mathrm{mod}\ m)\}$, with $a_i$ related to the canonical presentation of G given in Eq. (10). Denoting $\varphi(x)$ the Euler totient function, it holds $\{i^{\varphi(m)} : i = 1, \ldots, n\} \subset \mathcal{C}_G(C)$ for all integrals of the cyclic group $\mathbb{Z}_m$. In particular, the free group on T, contained in $\mathcal{C}_G(C)$, has n generators lying strictly within the nD hypercube (in the abelian support) with perimeter $\{\prod_{i=1}^n i^{s_i} : s_i = 0, m\}$.*

*Proof.* Clearly $\mathcal{C}_G(C) \geqslant C$ since $C$ is abelian. Thus, we can restrict the search of the other elements of $\mathcal{C}_G(C)$ within the Schreier transversal $T_n^{ab}$. We search for elements $d = \prod_{i=1}^n i^{n_i}$ that satisfy $c_{d\,c_{12}} = e$. Computing the l.h.s., using $c_{12}^{(j)} = c_{12}^{a_j}$ (for $j = 1, \ldots, n$) we obtain the equation $\prod_{i=1}^n a_i^{n_i} = 1\,(\mathrm{mod}\ m)$, thus obtaining the claimed set.

  Since each $a_k \in \mathbb{Z}_m^\times$ is coprime with $m$, by Euler's theorem the previous equation is satisfied, in particular, by setting all $n_i$ but one to 0, and the non-vanishing one to $\varphi(m)$.

  We observe that the free group on $T$ has $n$ generators. Indeed $Q \leqslant T \leqslant G^{ab}$, and $Q$ and $G^{ab}$ have the same rank and are free (cf. proof about the rank in Proposition 2.2). Since $\varphi(m) < m$, the full set $X = \{i^{\varphi(m)} : i = 1, \ldots, n\}$ can only be generated by elements lying strictly within the $n$-dimensional cube with perimeter $\{\prod_{i=1}^n i^{s_i} : s_i = 0, \varphi(m)\}$. Indeed, if there was a generator lying outside it could be shifted back inside the cube by multiplication of an element of $X$. $\qquad\square$

**Proposition 2.8.** *(Centralizer and flux periodicity)*

*Let $G$ be the CS-crystal with cyclic commutator group $C$. The subgroup $T$ of $\mathcal{C}_G(C)$, defined in Proposition 2.7, determines the periodicity of the flux configuration common to all irreps of $C$. To find the periodicity it is enough to identify the fluxes that equal the one at the origin.*

*Proof.* As shown in Proposition 2.7, $\mathcal{C}_G(C) = CT$, with $T \subseteq T_n^{ab}$. Consider the presentation of $G$ in its canonical form Eq. (10). For any dimension $n$, $c_{12}$ is the only generator of $C$ in $G$, when $C$ is cyclic.

Suppose the flux at position $g_{\mathbf{r}} \in T_n^{ab}$ equals the one at the origin $e$ in any of the Hamiltonian sectors associated to the representations of $C$. This means that the shifted loop $c_{12}^{(g_{\mathbf{r}})}$ is equal the parent one, $c_{12}$. So $g_{\mathbf{r}}$ commutes with $c_{12}$ and, as a consequence, with all elements of $C$, whence $g_{\mathbf{r}} \in T$. This commutation with all elements implies, in particular, that $g_{\mathbf{r}}$ determines the periodicity of the full pattern, not just of the loop at the origin.

At the same time any element of $T$ implies periodicity: if $t \in T$ and $g \in T_n^{ab}$ then $c_{12}^{(g\,t)} = c_{12}^{(g)}$, as can be seen using the identity $c_{12}^{(g\,t\,\bar{g})} = c_{12}^{(c_{\bar{g}\,\bar{t}}\,t)} = c_{12}^{(t)} = c_{12}$. $\qquad\qquad\square$

### E.3  Maximal integrals

**Proposition 3.1.** *(Existence of a maximal integral, simplified version from [38])*

*Let $C$ be a finite integrable group, $F_n$ a free group (for some $n$) and $F_n'$ its commutator group. For every integral $I \cong F_n/N$, with $N \lhd F_n$, it exists a group $G = F_n/N_P$, with $N_P = N \cap F_n'$, such that its commutator subgroup is isomorphic to $C$ and it exists $N_S = N/N_P \lhd G$ such that $I = G/N_S$.*

**Proposition 3.2.** *Given an integrable group $C$ and a maximal integral $G$, if $N \lhd G$ and $N \cap C$ is trivial then $N \leqslant Z$.*

*Proof.* Let $g \in G$ and $n \in N$. Normality implies $\bar{g}ng = n^{(g)} \in N$. Hence, we have $N \ni \bar{n}n^{(g)} = c_{ng} \in C$. But $N \cap C$ is trivial so we conclude $n^{(g)} = n$ for all $g \in G$. $\qquad\qquad\square$

**Proposition 3.3.** *(Subreps of $\mathcal{T}^L$ and primitive supercells reps $\mathcal{T}_{G/Y}^L$)*

*The representation $\check{\mathcal{T}}_\eta$ of $G/Y$ associated to $\tilde{\mathcal{T}}_\eta$, defined in Eq. (C.4), is equivalent to the left regular representation $\mathcal{T}_{G/Y}$.*

*Proof.* Let us denote by $T_\eta \subset L^2(G)$ the eigenspace associated to $\mathcal{T}_\eta$. $T_\eta$ contains functions that satisfy $\langle zg|\psi\rangle = \bar{\chi}_{\eta,g}^G \langle g|\psi\rangle$. Consider the map

$$U_\eta: \quad T_\eta \to L^2(G/Y)$$
$$|\psi\rangle \mapsto |\phi\rangle : \phi(gZ) = \chi_{\eta,g}^G \psi(g). \tag{E.19}$$

It is not hard to see that this map is unitary. The representation $\mathcal{T}_\eta'$ of $G/Y$ defined as $\mathcal{T}_\eta' = U_\eta \circ \check{\mathcal{T}}_\eta \circ U_\eta^\dagger$, with $^\dagger$ denoting the adjoint operation, is the left regular representation of $G/Y$. Indeed (we suppress the $\eta$ label here)

$$\langle gY|\mathcal{T}_{g'Y}'\phi\rangle = \langle gY|U \circ \check{\mathcal{T}}_{g'Y} \circ U^\dagger \phi\rangle$$
$$= \langle gY|U \circ \tilde{\mathcal{T}}_{g'} \circ U^\dagger \phi\rangle$$
$$= \chi_g^G \langle g|\tilde{\mathcal{T}}_{g'} \circ U^\dagger \phi\rangle$$
$$= \chi_g^G \bar{\chi}_{g'}^G \langle \bar{g}'g|U^\dagger \phi\rangle$$
$$= \chi_{\bar{g}'g}^G \langle \bar{g}'g|\psi\rangle$$
$$= \langle \bar{g}'gY|\phi\rangle. \tag{E.20}$$

$\qquad\qquad\square$

### E.4   2-dimensional CS-crystals with cyclic commutator group

**Corollary 3 of Theorem 1.5** (Straight unit CS-cell with two generators)

   *If $n = 2$ then $P = P_{\text{straight}}$ and the generator which is a power of $x_1$ must be in the center.*

*Proof.* Suppose $1^{m_1} 2^{m_2}$ is in a generating set of $P$. Then $1^m$ must be in the center since $[1^m, 2] = e$ and, as a consequence, $2^m \in \mathcal{C}_G(C)$ (we omit the index of the exponents when they are obvious). Moreover, since $1^m$ alone commutes with both 1 and 2 it must be in both $Z$ and $P$ (recall that $Z \leqslant \mathcal{C}_G(C)$). Thus also $2^m \in P$. It follows that it is possible to find generators of $P$ that are powers of generators of $H$. $\qquad\square$

**Remark.** *If $n > 2$ we might have $P > P_{\text{straight}}$. Suppose $n = 3$. There might exist powers $m_1 < V_{13}$ such that $c_{1^m,2} = e$ but $c_{1^m,3} \neq e$ such that $1^m \notin P_{\text{straight}}$. Still, nothing forbids that it might exist a power $m_2$ such that $1^m 2^m \in P$, hence this element would lie outside $P_{\text{straight}}$.*

**Proposition 4.1.** *(Presentation of Q)*

   *The group $Q$ of a 2-dimensional CS-crystal $G$ with cyclic commutator group $\mathbb{Z}_m$ is generated by the set $\{1^{n_1}, 2^{n_2}\}$ where the powers satisfy*

$$S_{n_i}(a_i) = 0 \ (\text{mod } m)$$
$$2 \leqslant n_i \leqslant m \tag{E.21}$$

*for $i = 1, 2$, here $S_n(a) = \sum_{s=0}^{n-1} a^s$ and the $a_i$'s are the integers determining $G$ appearing in Eq. (11).*

*Proof.* By Corollary E.4, since $Q < P_{\text{straight}}$ we must determine the powers $n_i$ such that $c_{i^{n_i} j} = e$, where $(i, j) = (1, 2), (2, 1)$. It is not difficult to see that

$$c_{i^n j} = \prod_{s=0}^{n-1} {}' c_{ij}^{(i^s)} = \prod_{s=0}^{'n-1} c_{ij}^{a_i^s} = c_{ij}^{\sum_{s=0}^{n_i-1} a_i^s} = c_{ij}^{S_n(a)} \quad (n \geqslant 1) \tag{E.22}$$

where we used the set of relators $R_a$, defined in Eq. (10). The exponent of $c_{ij}$ must be equal to a multiple of $m$, due to the first relator in Eq. (10). Thus, the first condition in Eq. (E.21) is obtained.

   To prove $2 \leqslant n_i \leqslant m$ we observe that, by pigeonhole principle, for any value $a$ there exist $C \geqslant 0$ and $N \geqslant 1$ such that $C + N \leqslant m$ and $S_{C+N}(a) = S_C(a) \ (\text{mod } m)$. The property above implies $a^C S_N(a) = 0 \ (\text{mod } m)$ thus it exists a positive integer $k$ such that ($S_N$ is a the geometric series)

$$a^C (a^N - 1)/(a - 1) = km.$$

Notice that $N \neq 1$ otherwise the latter equation would imply that $a$ is not coprime with $m$. The same properties imply ($C' > 0$)

$$S_{C+C'+N}(a) = S_{C+N}(a) + a^N a^C S_{C'}(a)$$
$$= S_{C+C'}(a) + (a^N - 1) a^C S_{C'}(a) \ (\text{mod } m). \tag{E.23}$$

The last term in the r.h.s is proportional to $m$, by the above equation, whence we get $S_{C+C'+N}(a) = S_{C+C'}(a) \ (\text{mod } m)$ for all positive $C'$. In other words, the sequence $\{S_r(a)\}_r$ is periodic with period $N \leqslant m$. In particular, $S_N(a) = 0 \ (\text{mod } m)$ with $2 \leqslant N \leqslant m$.

$\qquad\square$

**Remark.** *Equation Eq. (E.21) can be equivalently written as*

$$\begin{cases} n_i = m & \text{for } a_i = 1 \\ a^{n_i} = 1 & \text{for } a_i > 1 \end{cases} \tag{E.24}$$

**Proposition 4.2.** *(Presentation of Z) The center Z of a 2-dimensional CS-crystal G with cyclic commutator group $\mathbb{Z}_m$ is made of elements $z = 1^{n_1} 2^{n_2} c_{12}^{n_c}$ whose powers are in $\mathbb{Z}$ and satisfy*

$$\begin{cases} n_c(a_1 - 1) &= S_{n_2}(a_2) \\ n_c(a_2 - 1) &= S_{-n_1}(a_1) \end{cases} \pmod{m}. \tag{E.25}$$

*here the $a_i$'s are the integers determining G appearing in Eq. (11), $S_n(a) = (a^n - 1)/(a - 1)$ $(n \in \mathbb{Z})$ and the inverse integers $a^{-1}$ denotes the minimal integers $p \in \mathbb{N}$ such that $ap = 1 \pmod{m}$.*

*Proof.* It is enough to impose

$$c_{zi} = e \quad (i = 1, 2). \tag{E.26}$$

The identities $c_{1^{n_1} 2} = c_{12}^{S_{n_1}(a_1)}$ and $c_{1 2^{n_2}} = c_{12}^{S_{n_2}(a_2)}$, cf. Eq. (E.22), may come at hand in the computation. Here $S_r(a)$ can be extended to $r < 0$ using the identity $c_{\bar{i}j} = \bar{c}_{ij}^{(\bar{i})}$ to obtain $S_{r<0}(a) = -\sum_{s=1}^{|r|} a^{-s}$. Thus a definition that holds for any $r$ is

$$S_r(a) = (a^r - 1)/(a - 1) \tag{E.27}$$

where the case $a = 1$ is obtained from the limit.

Thus, Eq. (E.26) can be written as $c_{12}^{p_i} = e$ where $p_i$ is some power such that $p_i = 0 \pmod{m}$. Using modular arithmetic to simplify the expressions, we get

$$\begin{cases} n_c(1 - a_1)a_1^{n_1} a_2^{n_2} &= -S_{n_2}(a_2) \\ n_c(1 - a_2) &= S_{n_1}(a_1)a_2^{n_2} \end{cases} \pmod{m}, \tag{E.28}$$

Considering that $a_1^{n_1} a_2^{n_2}$ is congruent to 1, by $Z \leqslant \mathcal{C}_G(C)$ and Proposition 2.7, the above set of equations is equivalent to

$$\begin{cases} n_c(a_1 - 1) &= S_{n_2}(a_2) \\ n_c(a_2 - 1) &= -S_{n_1}(a_1)a_1^{-n_1} \end{cases} \pmod{m}. \tag{E.29}$$

Using $-S_{n_1}(a_1)a_1^{-n_1} = S_{-n_1}(a_1)$ one gets Eq. (E.25). □

**Remark.** *Notice that Eq. (E.21), specifying the elements of Q, is obtained from Eq. (E.25) with $n_c = 0$.*

**Corollary 4.3.** *Given a 2-dimensional CS-crystal G with cyclic commutator group $\mathbb{Z}_m$ if $\{1, 2\} \subset \mathcal{C}_G(C)$ (or equivalently $a_1 = a_2 = 1$) then $P \geqslant \widetilde{Z} = \{1^m, 2^m\}$.*

**Corollary 4.4.** *Given a 2-dimensional CS-crystal G with cyclic commutator group $\mathbb{Z}_m$ if $\{1, 2\} \subset \mathcal{C}_G(C)$ (or equivalently $a_1 = a_2 = 1$) then $Z_{CG} = G$. Otherwise, $Z_{CG}$ is nontrivial if only if l.c.d.$\{a_1 - 1, a_2 - 1, m\} \neq 1$.*

*Proof.* We prove the last statement. If $Z_{CG}$ is nontrivial if and only if we can set $n_1 = n_2 = 0$ in Eq. (E.25) so that $0 \neq n_c \propto m/(a_1 - 1) \propto m/(a_2 - 1)$.

It is clear that if l.c.d.$\{a_1 - 1, a_2 - 1, m\} = 1$ then $n_c = 0 \pmod{m}$ is the only integer solution. Conversely, if l.c.d.$\{a_1 - 1, a_2 - 1, m\} = s \neq 1$, then we seek integer values of $k_x$ such that $n_c = k_1 l/r_1 = k_2 l/r_2$ where $l = m/s$ and $r_x = (a_x - 1)/s$ $(x = 1, 2)$ are integers. Clearly $k_x = r_x$ is a valid solution, as it implies $n_c = l < m$. □

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
