# Peer review of "Translation Groups for arbitrary Gauge Fields in Synthetic Crystals with real hopping amplitudes"

_SciPost Physics_

## Round 2 · Referee Report · Anonymous (Referee 1) · 2025-10-29

Report

In the paper entitled "Translation Groups for arbitrary Gauge Fields in Synthetic Crystals with real hopping amplitudes", the author has introduced and studied the Cayley-Schreier (CS) lattices, in particular, by showing that these systems naturally realize the generalization of the magnetic translation groups to arbitrary discrete gauge groups G. By focusing on the Abelian cases, the author has shown the Bloch decomposition and corresponding Bloch Hamiltonian eigenstates (related to the generalized magnetic translations) for some concrete lattice systems.

I find the paper very interesting with several original results although the style of the paper is very technical and not of easy access for condensed matter physicists who are not very familiar with advanced mathematical formalism of the paper such as advanced group theory, Peter-Weil theorem and Cayley graphs.

Honestly, I would have expected a deeper and more detailed discussion about generalized Bloch theorem for the CS lattices, somehow a more explicit presentation, similar to what has been done in some works on hyperbolic lattices, see for instance Refs [21,22] cited by the author. I believe it would be beneficial for the readers if this point is presented and discussed in a more physics-friendly way instead of the very formal presentation by the author concerning Bloch decomposition.

More importantly, I have two questions for the author:

1) For the examples discussed in section 9, the author explicitly discusses the spectra which result to be gapless as shown in Fig.6. On the other hand, it is well known that magnetic translations are key ingredient that ensure the topological stability and gapped nature of the Quantum Hall states for instance. The situation changes when we consider different types of crystalline solids, especially those with more complex symmetries. In these materials, magnetic translations (or generalizations of them) can conspire with other crystal symmetries to protect gapless points in the electronic band structure. How is the situation in the case of CS lattices? Is there any way to understand if a given model, related to a group G, supports or not a well-defined gap before doing brute-force calculations? What is the role of discrete symmetries in this context?

2) How the topology of the Cayley graphs influence the physical features of the CS lattice models such as their generalized magnetic translations? It is well known that Cayley graphs can possess non-trivial topology, see for instance the following work:

Mark Brittenham, Susan Hermiller, Derek Holt, Algorithms and topology of Cayley graphs for groups, Journal of Algebra 415, 112 (2014).

Thus, it would be interesting and relevant if the author could comment the role of graph topology and its possible physical implications for the CS lattices (although it may be highly non-trivial to get a complete picture at this stage).

Overall, I support the publication of this work on SciPost once the author addresses the points I raised above.

Recommendation

Publish (meets expectations and criteria for this Journal)

---

## Editorial Decision

in_refereeing